# Unbiased Gradient Low-Rank Projection

## Abstract

Memory-efficient optimization is critical for training increasingly large language models (LLMs). A popular strategy involves gradient low-rank projection, storing only the projected optimizer states, with GaLore being a representative example. However, a significant drawback of many such methods is their lack of convergence guarantees, as various low-rank projection approaches introduce inherent biases at each step relative to the original optimization algorithms even after taking expectation over stochastic sampling, which contribute to performance gaps compared to full-parameter training. Aiming to tackle this problem, this paper investigates the layerwise sampling technique for debiasing low-rank projection mechanisms. In particular, an instantiation of the paradigm gives rise to a novel and step-wise unbiased low-rank optimization method built upon GaLore's mechanism and the Muon algorithm, named GaLore Unbiased with Muon (GUM). We theoretically prove our method matches the convergence guarantees of the base Muon algorithm while preserving the memory efficiency of low-rank techniques. Empirical experiments on LLM fine-tuning and pretraining also demonstrate non-trivial improvements over GaLore and even better performance than full-parameter training. Further investigation shows that the improvement of this technique comes from a more uniform distribution of knowledge inside layers, leading to more efficient utilization of the model parameter space and better memorization.

## 1 Introduction

Large language models (LLMs) have demonstrated impressive performance across a diverse range of tasks, including conversation (Ouyang et al., 2022; Grattafiori et al., 2024b), mathematical reasoning (Guo et al., 2025), and agentic applications (Qin et al., 2025). The advancement of these powerful LLMs demands substantial GPU memory due to the large size of the underlying models. For example, training a 70B model with full parameters requires approximately 1.2 terabytes of GPU memory, which exceeds the capacity of even 8×H100 GPUs.

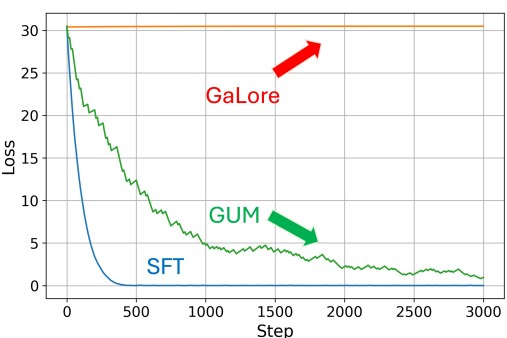

Figure 1: A counterexample of GaLore in linear regression with Muon optimizer (Jordan et al., 2024), where its debiased version GUM converges while GaLore fails to converge.

To address this issue, memory-efficient training techniques such as GaLore (Zhao et al., 2024) have been introduced. GaLore projects gradients into a low-rank space, reducing the memory footprint of optimizer states during training. Specifically, it employs the top-$r$ components from Singular Value Decomposition (SVD) to define a compact low-rank space, into which the gradients are projected as $R_t \leftarrow P_t^\top G_t$. The optimization step is then performed in this low-rank space, enabling memory savings for the optimizer states. For example, the first and second moments in Adam (Kingma & Ba, 2014) are updated using $\tilde{M}_t \leftarrow \beta_1 \tilde{M}_{t-1} + (1 - \beta_1)R_t$ and $\tilde{V}_t \leftarrow \beta_2 \tilde{V}_{t-1} + (1 - \beta_2)R_t$, where the low-rank projected gradient $R_t$ replaces the original gradient $G_t$. After the optimization step, the parameter update is projected back to the original space.

---

**Algorithm 1** Low-rank projection based gradient descent algorithms

---

1: **Input:** Initial weight $W_0 \in \mathbb{R}^{m \times n}$ (suppose $m \leq n$), number of iterations $K$, learning rate $\eta$, projection rank $r$.
2: **for** $t = 0$ **to** $K - 1$ **do**
3:      $G_t = G(W_t)$                                       ▷ Obtain the gradient at $W_t$
4:      $P_t \leftarrow \texttt{get\_projector()}$                        ▷ Obtain the projector $P_t \in \mathbb{R}^{m \times r}$
5:      $\tilde{G}_t = P_t^\top G_t$                ▷ Obtain projection of the gradient in the low-rank space
6:      $S_t \leftarrow \texttt{optimizer.update\_state}(\tilde{G}_t)$         ▷ Run the base algorithm with $\tilde{G}_t$
7:      $W_{t+1} = W_t - \eta P_t S_t$          ▷ Project the update back and update the weights
8: **end for**

---

Nevertheless, most low-rank optimization methods introduce biased gradient estimations at each step during training (Muhamed et al., 2024; Zhang et al., 2024a; He et al., 2024; Huang et al., 2025), which can lead to suboptimal convergence behavior and measurable performance gaps compared to standard full-parameter training. These biases arise because low-rank projections, while computationally and memory efficient, do not fully preserve the direction and magnitude of the true gradient, especially in high-dimensional parameter spaces. As a result, the optimization trajectory diverges from that of full-precision training, potentially causing slower convergence, reduced final model quality, or instability in certain regimes (Zhao et al., 2024; Ding et al., 2022; Zhang et al., 2024a; Huang et al., 2025). This limitation is particularly critical when pre-training large language models (LLMs), where even small discrepancies in gradient estimation can propagate and amplify across many layers and iterations.

To address this fundamental issue, we investigate the general debiasing technique using layerwise sampling (Pan et al., 2024), which preserves the memory efficiency of training methods via randomly freezing most of the layers. Specifically, the unique strength of layerwise sampling over the typical low-rank projected algorithms of GaLore is analyzed both theoretically and empirically. The introduction of the debiasing technique into GaLore gives rise to a new algorithm called **Galore Unbiased with Muon (GUM)**, which demonstrates much better convergence guarantees and practical performance in LLM training tasks. We summarize our major contributions as follows:

- We investigate the layerwise-sampling debiasing technique and propose a novel algorithm called **G**aLore **U**nbiased with **M**uon (**GUM**), which unifies the strengths of GaLore and Muon. GUM achieves the same theoretical convergence guarantees as Muon while retaining the memory efficiency of GaLore, enabling scalable and effective training of large models.

- Empirical experiments in LLM training demonstrate that GUM consistently outperforms GaLore in instruction-following, mathematical reasoning, and commonsense reasoning tasks under the same memory budget. Surprisingly, in LLM pre-training experiments, GUM even outperforms full-parameter trained AdamW by a non-trivial overall accuracy margin of 0.3%-1.1%, while obtaining on-par or better performance than AdamW in 6 out of 7 tasks.

- We analyze the underlying reasoning of GUM's empirical improvements, discovering that its high-rank update nature leads to a larger overall stable rank and more evenly distributed singular values in model weights, which further induce a more long-tailed activation pattern in trained models. This implies the performance gain is brought by more efficient utilization of the model parameter space, in other words, better memorization.

## 2 RELATED WORK

**Parameter-Efficient Algorithms in Practice.** Parameter-Efficient Fine-Tuning (PEFT) methods are widely adopted for training large-scale LLMs in practice. A typical approach is LoRA (Hu et al., 2022), which freezes the original model and attaches a small trainable low-rank adapter, thereby reducing memory consumption and improving training efficiency. However, LoRA has been reported to exhibit a non-trivial performance gap compared to full-parameter training (Ding et al., 2022; Lialin et al., 2023), due to its altered parameter space. These changes in parameter space also introduce theoretical challenges in analyzing LoRA's convergence properties with respect to the original parameter space. To address the aforementioned deficiencies and extend LoRA to

larger-scale training settings, GaLore (Zhao et al., 2024) proposes a different approach, which projects the gradients—rather than the parameters—into low-rank spaces. In doing so, the error between the full gradients and the approximated gradients becomes numerically quantifiable, as they now operate within the same parameter space.

Following GaLore, a number of low-rank projection-based algorithms have emerged, where the key component follows a similar paradigm to Algorithm 1, but with different projection matrices $P$. GaLore utilizes the top-$r$ entries $U[:,:r]$ from SVD, which is computationally expensive. To address this issue, GRASS (Muhamed et al., 2024) derives a sparse projection matrix $P$ based solely on the row norms of the gradients. Specifically, each projection entry is sampled from a multimodal distribution proportional to the row norms. GRASS has been reported to achieve performance comparable to GaLore with lower computational cost, though no theoretical guarantees have been provided regarding its convergence. LoGE (Zhang et al., 2024a) obtains the low-rank projection $P$ by decomposing the original weight matrix $W = BC$, thereby implicitly allowing the backward gradient to be low-rank. However, it is difficult to guarantee theoretical convergence due to the empirical nature of the low-rank decomposition. GradNormLoRP (Huang et al., 2025) combines ideas from LoRA and GaLore, resulting in a two-level projection $P$ that further enhances memory efficiency and reduces training cost. A variety of salience-aware sparse projections are also employed in (Guo et al., 2020; Sung et al., 2021; Ansell et al., 2021; Das et al., 2023; Liu et al., 2024a), each using different saliency metrics.

Despite the strong empirical performance across various practical settings, most of the aforementioned methods lack guarantees regarding their theoretical convergence rates, which can be attributed to the biasedness of the projected gradients. To bridge this gap, we investigate the debiasing technique of layerwise-sampling that compensates for the errors introduced by low-rank projected updates, aiming to improve their theoretical convergence guarantees while maintaining practical memory efficiency.

**Unbiased Optimization Methods.** The research on unbiased methods is an important part of the optimization field, especially for distributed and memory-efficient optimization. This includes methods of unbiased quantization (Alistarh et al., 2017; Suresh et al., 2017; Wang et al., 2022) and unbiased sparsification (Wangni et al., 2018; Stich et al., 2018; Wang et al., 2018). The unbiased property of these methods enables low communication/memory burden while maintaining guaranteed convergence. For the recently popular low-rank projection-based methods, Fira (Chen et al., 2024) provides an attempt to involve full-rank information by adding a scaled gradient projection to the update, but without a rigorous theoretical justification of the approach. GoLore (He et al., 2024) is probably the closest to building an unbiased algorithm. However, they employ a totally random projection matrix for the algorithm to enable the convergence guarantee, which may fail to capture the loss landscape properties and lead to slow convergence.

**Muon Optimizer.** Muon (Jordan et al., 2024) is a novel optimizer proposed recently, which is gaining rapidly increasing attention because of its great potential in training large foundation models (Liu et al., 2025a; Kimi, 2025), empirically outperforming AdamW on specific large-scale tasks. On the theoretical side, An et al. (2025); Li & Hong (2025) proves its non-convex deterministic and stochastic convergence, respectively, showing a strong theoretical guarantee for the optimizer.

## 3 ALGORITHM

### 3.1 GALORE UNBIASED WITH MUON

As previously shown in Algorithm 1, the core of low-rank gradient methods is to only store the low-rank projected optimizer states, i.e., related to $\tilde{G}_t \in \mathbb{R}^{m \times r}$, which is then projected back to the weight space by multiplying $P_t$ to update the weight $W_t$. The update conceptually shares similarities with running the base optimizer using low-rank projected gradients $P_t P_t^\top G_t$ instead of $G_t$.

This inspires the key idea of *debiasing*, that is, to compensate for biased errors introduced by the low-rank projection $P_t P_t^\top G_t$. To implement this while retaining memory efficiency, we refer to the main idea of LISA (Pan et al., 2024), which allows some of the blocks to be sampled uniformly with probability $q$ in each period. This compensated full-rank updates use $G_t - P_t P_t^\top G_t$, while other blocks still do the original low-rank update. By carefully balancing the scaling constants for the two different updates, the biased low-rank term can be canceled out in expectation, resulting in an

---

**Algorithm 2** **G**aLore **U**nbiased with **M**uon (**GUM**)

---

1: **Input:** $\{W_{0,\ell} \in \mathbb{R}^{m_\ell \times n_\ell}\}$ with each $\ell$ corresponding to the $\ell$-th block of parameters, number of blocks $N_L$, sampling period $K$, rank $r$ for each layer, full-rank update layer number $\gamma$

2: **for** $t = 0$ **to** $T/K - 1$ **do**

3:     **for** $\ell = 1$ **to** $N_L$ **do**

4:         Initialize $R_{t,0,\ell} = 0$                       $\triangleright$ Restart the momentum to clear memory

5:         $G_{t,0,\ell} = G(W_{t,\ell})$                 $\triangleright$ Obtain the gradient of the $\ell$-th layer at $W_t$

6:         $U_{t,\ell}, S_{t,\ell}, V_{t,\ell} = \texttt{SVD}(G_{t,0,\ell})$       $\triangleright$ Compute SVD of gradient obtained at $W_{tK}$

7:         $P_{t,\ell} = U_{t,\ell}[:,:r]$          $\triangleright$ Obtain GaLore projector $P_{t,\ell} \in \mathbb{R}^{m \times r}$ (suppose $m_\ell \leq n_\ell$)

8:     **end for**

9:     Each block $\ell$ is sampled to do full-rank updates with probability $q_{t,\ell} \equiv q = \frac{\gamma}{N_L}$

10:     **for** $k = 0$ **to** $K - 1$ **do**

11:         Run (1) for all blocks sampled to compute low-rank update

12:         Run (2) for all blocks sampled to compute full-rank update

13:     **end for**

14: **end for**

---

unbiased estimation of gradients across iterations. Due to page limit, we present this general unbiased algorithm paradigm in Algorithm 3 in Appendix A.

For a practical instance of this paradigm, we consider applying GaLore as the low-rank projection method and Muon as the base algorithm, which gives birth to our proposed optimization algorithm, called **G**aLore **U**nbiased with **M**uon (**GUM**), as presented in Algorithm 2.

In one training process, the algorithm contains separated periods just like the vanilla GaLore and LISA. During each period $t$, each block of parameters is sampled to do full-rank updates with probability $q_{t,\ell}$. In each iteration $k$ in the period, we first compute the projection matrix $P_{t,k,\ell}$ and sample the layers to do full-rank updates in this period.

If block $\ell$ is sampled to do the low-rank update, we apply the following update adapted from Muon with $G_{t,k,\ell} = G(W_{tK+k,\ell})$ as the gradient of block $\ell$ at iteration $k$ in period $t$:

$$R_{t,k,\ell} = \beta R_{t,k-1,\ell} + \frac{1}{1 - q_{t,\ell}} P_{t,\ell}^\top G_{t,k,\ell}$$

$$W_{Kt+k+1,\ell} = W_{Kt+k,\ell} + \eta_{t,k} P_{t,\ell} \texttt{NewtonSchulz}(R_{t,k,\ell}) \tag{1}$$

Note that if we set $q_{t,\ell} = 0$, (1) is exactly GaLore with Muon as the base optimizer, which we will refer to as GaLore-Muon. In terms of memory consumption, we can see that the optimizer states requiring storage are the projection matrix $P_{t,\ell} \in \mathbb{R}^{m_\ell \times r}$ and $R_{t,k,\ell} \in \mathbb{R}^{r \times n_\ell}$. Otherwise, the block is sampled to compute high-rank updates, and the compensated projection update is applied.

$$R_{t,k,\ell} = \beta R_{t,k-1,\ell} + \frac{1}{q_{t,\ell}} \left( G_{t,k,\ell} - P_{t,\ell} P_{t,\ell}^\top G_{t,k,\ell} \right)$$

$$W_{tK+k+1,\ell} = W_{tK+k,\ell} + \eta_{t,k} \texttt{NewtonSchulz}(R_{t,k,\ell}) \tag{2}$$

In this case, $P_{t,\ell} \in \mathbb{R}^{m_\ell \times r}$ and $R_{t,k,\ell} \in \mathbb{R}^{m_\ell \times n_\ell}$ are required to be stored.

Summarizing both cases, the overall memory consumption comparison with the vanilla GaLore-Muon algorithm is obtained, as shown in Table 3. The memory consumption of GUM is higher than that of GaLore when using the same projection rank $r$, due to the use of probabilistic full-rank updates. However, as demonstrated in Section 5, by employing a smaller projection rank $r'$ as a trade-off, the benefits of this additional memory consumption are sufficient to recover the performance loss and even achieve a smaller overall memory footprint.

We can show that this update is unbiased compared to the original Muon update.

**Lemma 1** (GUM is unbiased). *A single iteration of Algorithm 2 for $W \in \mathbb{R}^{m \times n}$ is equivalent to*

$$\tilde{M}^+ = \beta \tilde{M} + \tilde{G}$$

$$W^+ = W - \eta \texttt{NewtonSchulz}(\tilde{M}^+)$$

*with $\mathbb{E}[\tilde{G}] = G \in \mathbb{R}^{m \times n}$, where $G$ denotes the gradient obtained at $W$, and the expectation is taken over stochastic sampling and layerwise sampling random variable $\zeta$, where*

- $\zeta = 0$ *when layer $\ell$ does a low-rank update, with probability $1 - q$,*

- $\zeta = 1$ *when layer $\ell$ does a full-rank update, with probability $q$.*

This unbiased technique is crucial for the convergence of the algorithm. As we will see in the next subsection, GUM can recover similar convergence properties as the original Muon algorithm, regardless of the employed projection matrix. This demonstrates substantial theoretical advantages over the original biased GaLore-Muon algorithm.

## 4 CONVERGENCE ANALYSIS OF GUM

In this section, we present the convergence analysis of GUM. We consider the following assumptions for the minimization problem $\min_{W \in \mathbb{R}^{m \times n}} f(W)$ with $m \leq n$.

**Assumption 1** (Lower bounded). *There exists $f^* > -\infty$ such that $f(W) \geq f^*$ for all $W \in \mathbb{R}^{m \times n}$.*

**Assumption 2** (Smoothness). *$f$ is $L_{\mathrm{op}}$-smooth with respect to the spectral norm $\|\cdot\|_{\mathrm{op}}$, i.e.,*

$$\|\nabla f(W_1) - \nabla f(W_2)\|_* \leq L_{\mathrm{op}} \|W_1 - W_2\|_{\mathrm{op}},$$

*for all $W_1, W_2 \in \mathbb{R}^{m \times n}$. $\|\cdot\|_{\mathrm{op}}$ and $\|\cdot\|_*$ denotes the spectral norm and trace norm respectively.*

**Assumption 3** (Gradient noise). *We assume the stochastic gradient $G(W)$ obtained at $W$ is unbiased and there exists a matrix $V \in \mathbb{R}^{m \times n}$ such that*

$$\mathbb{E}[N(W)] = 0 \quad \text{and} \quad \mathbb{E}\left[N(W)N(W)^\top\right] \preceq VV^\top,$$

*where $N(W) \triangleq G(W) - \nabla f(W)$ and $A \preceq B$ denotes that $B - A$ is positive semidefinite.*

Assumption 1 is standard in non-convex analysis. Based on the equivalence between norms, Assumption 2 implies nothing more than the standard smoothness assumption on Frobenius norm, but is more suitable in analyzing GUM or Muon (Jordan et al., 2024). Assumption 3 can also imply the standard bounded variance assumption by $\mathbb{E}[\|N(W)\|_{\mathrm{F}}^2] \leq \|V\|_{\mathrm{F}}^2$. The style of these assumptions can be found in previous work on analyzing adaptive methods and Sign-based methods (Bernstein et al., 2018; Crawshaw et al., 2022; Liu et al., 2024b; An et al., 2025), where the assumptions are employed for more fine-grained analysis and analyzing the potential benefits of these optimizers.

**Assumption 4** (Exact Newton Schulz). *We consider the case where the Newton-Schulz iteration computes the exact solution, i.e., `NewtonSchulz`$(X) = UV^\top$ with $X = U\Sigma V^\top$ as the SVD of $X$.*

Assumption 4 is needed for analyzing Muon. As noted in Jordan et al. (2024); Liu et al. (2025a), though the Newton-Schulz iteration adopted in Muon does not compute the exact $UV^\top$ matrix, it turns out that this error has little influence on the training curve. Then, based on the assumptions, we can obtain the convergence guarantee for GUM.

**Theorem 1** (Non-convex Convergence). *Under Assumption 1-4, after running a total of $T$ iterations for Algorithm 2 with parameters set as (12), it holds that*

$$\min_{0 \leq s \leq T-1} \mathbb{E}\left[\|\nabla f(W_s)\|\right] \leq \mathcal{O}\left(\frac{1}{\alpha}\sqrt{\frac{L_{\mathrm{op}}\Delta}{T}} + \left(\frac{L_{\mathrm{op}}\Delta \|V\|_*^2}{\alpha^5 T}\right)^{\frac{1}{4}} + \frac{\|V\|_*}{\sqrt{\alpha^3 T}}\right),$$

*where $\Delta \triangleq f(W_0) - f^*$ and $\alpha \triangleq \min\{q, 1 - q\}$.*

The proof can be found in Appendix B. The convergence theorem for GUM leads to several important observations. Firstly, when we set $q$ to be an absolute constant, the convergence of GUM matches exactly the convergence rate of Muon. In the deterministic case, it matches the convergence result of Muon proven in An et al. (2025). When the noise $V$ is the dominant term, it also matches the $\mathcal{O}(T^{-1/4})$ rate proven in Li & Hong (2025). Moreover, since we use more fine-grained and appropriate assumptions to analyze GUM, Theorem 1 shows an even better dimensional dependence than Li & Hong (2025). This consistency shows the power of the unbiased design, maintaining the memory reduction of gradient low-rank methods without sacrificing the convergence guarantee.

**Remark 1.** *In Theorem 1, the optimal choice is $q = 0.5$ for fast convergence. Although any positive constant $q \in (0, 1)$ ensures the proved theoretical convergence, in practical settings, a large constant $q$ may lead to huge memory consumption. This indicates a fundamental tradeoff between time and space, which is controlled by the choice of $q$. If the memory requirement is constant, it is preferable to choose the largest affordable $q \leq 0.5$. If the memory support is dynamic, e.g., the number of nodes decreases at a later stage of training, it is better to schedule a diminishing $q$ to adapt to this dynamic memory.*

As noted by He et al. (2024), GaLore using SGD with momentum (SGDM) as the base algorithm converges in the deterministic non-convex setting, but can possibly diverge when the gradient noise is large. We also empirically examine an extreme counterexample where GaLore-Muon doesn't converge at all in Section 5. Clearly, GUM fixes this problem. GoLore (He et al., 2024) is also designed to correct the convergence of GaLore. However, though GoLore shows a good convergence guarantee when the base algorithm is SGDM, it employs a thoroughly random projection matrix to do low-rank updates, failing to capture the potential gradient low-rank properties as the GaLore projection matrix does. This can lead to a much slower convergence speed when applied to real training tasks.

## 5 EXPERIMENTAL RESULTS

### 5.1 SYNTHETIC SETTINGS

To better illustrate how GaLore may fail due to the low-rank projection, we consider the following synthetic noisy problem.

**Setup.** The settings of the experiment are generally the same as the synthetic experiment in He et al. (2024). We consider the following noisy linear regression problem.

$$\min_{X \in \mathbb{R}^{n \times n}} f(x) \triangleq \frac{1}{2} \|AX\|_{\mathrm{F}}^2 + \langle B, X \rangle, \quad \nabla f(X; \xi) = \nabla f(X) + \xi \sigma C,$$

where $A = [I_{n-r} \quad 0] \in \mathbb{R}^{(n-r) \times n}$, $B = \begin{bmatrix} D & 0 \\ 0 & 0 \end{bmatrix} \in \mathbb{R}^{n \times n}$ with $D \in \mathbb{R}^{(n-r) \times (n-r)}$ a Gaussian random matrix, $C = \begin{bmatrix} 0 & 0 \\ 0 & I_r \end{bmatrix} \in \mathbb{R}^{n \times n}$, $\xi$ is a random variable with probability $0.5$ to be $1$ and probability $0.5$ to be $0$, and $\sigma$ is a constant controlling the noise level. It is straightforward to verify that this is a smooth and convex optimization problem, with bounded gradient variance.

In our experiment, we specifically set $n = 20$, $r = 12$, $\sigma = 100$ to construct a small-scale but noisy problem. For the vanilla (biased) GaLore Muon algorithm, we set the projection rank to be 12 as well. For GUM, we set $r = 2$ and $q_{t,\ell} = 0.5$. We can see that in this case, the memory footprints of the two algorithms are the same.

**Results.** The convergence result is shown in Figure 1. We adjust the minimum loss to 0 to better visualize the difference. As we can see, GaLore fails to converge at all, while GUM converges to a comparable accuracy with the full-parameter Muon baseline. The experiment shows a clear benefit of the unbiased method, at least in noisy settings.

Here is a more detailed analysis of why these conditions lead to GaLore's failure. In this synthetic problem, the noise level is set to be large and has rank $r = 12$, which is equal to the projection rank of GaLore. Since the noise is in a dominant position, every time the $r$ largest sin-

Table 1: **Space complexity** comparison between GaLore and GUM for a block $W \in \mathbb{R}^{m \times m}$ with $r' < r \leq m$ respectively. GUM uses a full-rank update with probability $q \in [0, 1]$, where the memory GUM has the same memory consumption when $q = 2(r - r')/(m - r')$.

| Method | Space Complexity |
|--------|------------------|
| GaLore | $\mathcal{O}(2mr)$ |
| GUM | $\mathcal{O}((2 - q)mr' + qm^2)$ |
| SFT | $\mathcal{O}(m^2)$ |

gular values of the stochastic gradient $\nabla f(X; \xi)$ come from the noise, so do the corresponding singular vectors and the GaLore projection matrix. This meaningless projection makes the training process not even take a single effective step towards solving the problem. Therefore, this synthetic

experiment shows an extreme case in which GaLore can fail when the gradient noise is large. Also, the experiment shows that GUM fixes the non-convergence problem with the same memory cost as GaLore-Muon.

## 5.2 LLM FINE-TUNING SETTINGS

To verify the empirical effectiveness of the proposed algorithm in practice, we compare GUM with GaLore in LLM fine-tuning settings.

**Setup.** The performance of the fine-tuned models is evaluated on two types of tasks: 1) IFEval (Zhou et al., 2023), an instruction-following benchmark that assesses models' adherence to explicit, verifiable instructions, and 2) GSM8K (Cobbe et al., 2021a), a mathematical reasoning benchmark that evaluates models' problem-solving skills in grad-school level math questions.

For model choices, LLaMA3-8B (Grattafiori et al., 2024a), Qwen2.5-7B (Qwen et al., 2025), and Gemma2-9B (Team et al., 2024) are adopted, which are commonly used in practical applications.

For training datasets, GPT-4-LLM is adopted on the instruction-following tasks of IFEval, which consists of 54.6K high-quality GPT-4-generated instruction-response pairs across various instruction categories. As for the mathematical reasoning task of GSM8K, a 2K-sized high-quality mixture [1] from DART-Math (Tong et al., 2024), Ultra-Interact (Yuan et al., 2024), MathInstruct (Yue et al., 2023), and Orca-Math (Mitra et al., 2024) is employed, which allows strong models such as Qwen-2.5-7B to still obtain reasonable improvements after fine-tuning.

For hyperparameters, we adopt a rank of 512 for GaLore and 2 + 128 for GUM. The baselines include Full-parameter Training with Muon (Jordan et al., 2024) (FT-Muon), Full-parameter Training with AdamW (Loshchilov & Hutter, 2019) (FT-AdamW), Gradient Low-Rank Projection (GaLore) (Zhao et al., 2024), Fira (Chen et al., 2024), Golore (He et al., 2024), LDAdam (Robert et al., 2025), and Apollo (Zhu et al., 2025), where further details are available in Appendix C.

**Memory Efficiency.** We conducted peak GPU memory experiments to evaluate GUM's memory efficiency, demonstrating its comparable or reduced memory footprint relative to GaLore. Specifically, we focus on two key hyperparameters: the rank and the number of selected layers for full-rank updates in GUM. To ensure a fair comparison, all methods used a consistent mini-batch size of 1, without employing additional GPU memory-saving techniques such as offloading (Ren et al., 2021) or flash attention (Dao et al., 2022; Dao, 2024).

As shown in Table 3, the GUM configuration reaches comparable or better memory consumption than GaLore. This improvement is not limited to a single case; consistent memory savings are observed across multiple model architectures.

Table 3: **Peak GPU memory usage** across different model architectures and configurations, emphasizing the variations among them. As specified in the table, the GUM configuration 2 + 128 involves updating two layers with full-rank gradients, while all other layers are updated with low-rank gradients of rank $r = 128$.

| Model | GaLore | GUM Layers + Rank | |
|---|---|---|---|
| | 512 | 4 + 128 | 2 + 128 |
| LLaMA-3-8B | 42G | 41G | **40G** |
| Qwen-2.5-7B | 41G | 40G | **39G** |
| Gemma-2-9B | 47G | 46G | **44G** |

**Results.** As shown in Table 2, GUM consistently outperforms GaLore in both tasks, highlighting its robustness and general effectiveness.

A closer look at GSM8K results reveals that GUM achieves notable improvements and even outperforms full-parameter training methods, suggesting its strength in enhancing reasoning capabilities. In Section 5.4, it will be revealed that this improvement is very likely to have originated from its improvements in memorization, especially when the learned activations are required to be long-tailed.

---

[1] The dataset is from `https://huggingface.co/datasets/HanningZhang/scalebio_distill_qwen_math`, generated using the same setting as Appendix A.2 of (Pan et al., 2025).

Table 2: **LLM Fine-tuning Results.** Trained models are evaluated on IFEval (instruction-following) and GSM8K (mathematical reasoning). All experiments are conducted on a single H100 GPU.

| Model | Memory Efficient | Method | IFEval | | GSM8K |
| --- | --- | --- | --- | --- | --- |
| | | | Prompt-level Strict-Accuracy | Prompt-level Loose-Accuracy | Accuracy |
| LLAMA-3-8B | ✗ | FT-AdamW | 23.66 | 25.14 | 57.39 |
| | | FT-Muon | 23.11 | 26.06 | 57.65 |
| | ✓ | Apollo (Zhu et al., 2025) | 19.04 | 21.63 | 56.03 |
| | | GaLore (Zhao et al., 2024) | 21.07 | 22.74 | 57.38 |
| | | Fira (Chen et al., 2024) | 21.81 | 23.73 | 56.41 |
| | | LDAdam (Robert et al., 2025) | 22.74 | 24.40 | 57.92 |
| | | GoLore (He et al., 2024) | **23.01** | **24.95** | 57.54 |
| | | GUM | 22.37 | 24.03 | **58.45** |
| QWEN-2.5-7B | ✗ | FT-AdamW | 35.12 | 39.74 | 85.75 |
| | | FT-Muon | 34.38 | 39.19 | 85.90 |
| | ✓ | Apollo | 31.61 | 36.41 | 85.67 |
| | | GaLore | 33.09 | 37.71 | 86.28 |
| | | Fira | 32.35 | 36.04 | 86.81 |
| | | LDAdam | 28.10 | 30.31 | 83.78 |
| | | GoLore | 30.87 | 35.67 | 86.66 |
| | | GUM | **33.46** | **38.82** | **86.81** |
| GEMMA-2-9B | ✗ | FT-AdamW | OOM | OOM | OOM |
| | | FT-Muon | 28.47 | 32.16 | 76.92 |
| | ✓ | Apollo | 25.14 | 28.10 | 75.28 |
| | | GaLore | 30.31 | 33.64 | 77.18 |
| | | Fira | 29.21 | 33.64 | 75.44 |
| | | LDAdam | 28.84 | 32.53 | 75.13 |
| | | GoLore | 31.05 | 34.38 | 74.98 |
| | | GUM | **33.27** | **36.60** | **77.48** |

## 5.3 LLM PRE-TRAINING SETTINGS

To provide stronger evidence for validating the effectiveness of GUM, a standard pre-training setting is introduced to compare different training methods' performance.

**Setup.** To evaluate the improvements in commonsense reasoning, the following downstream tasks are employed: ARC (Clark et al., 2018), OpenBookQA (Mihaylov et al., 2018b), HellaSwag Zellers et al. (2019a), PIQA (Bisk et al., 2020), SIQA (Sap et al., 2019), and WinoGrande (Sakaguchi et al., 2021a), which are common choices for LLM pre-training (Hoffmann et al., 2022a; Groeneveld et al., 2024; Zhang et al., 2024b). For model choice, following the standard setting in Zhao et al. (2024), the experiments covered three model sizes—60M, 130M, and 350M parameters of LLaMA. For training datasets, we employ the widely-used C4 corpus (Raffel et al., 2023) under configurations guided by the Chinchilla scaling law (Hoffmann et al., 2022b): 1.5B tokens for 60M, 2B tokens for 130M, and 7B tokens for 350M. For baselines, in addition to Galore and full-parameter training methods, we include Fira (Chen et al., 2024) and Subtrack++ (He et al., 2024). Further details are available in Appendix C.3.

**Results.** The performance comparison presented in Table 4 clearly indicates that GUM achieves consistently better results than GaLore and, more surprisingly, even full-parameter training methods like AdamW and Muon. This improvement can largely be attributed to the unbiased low-rank update mechanism employed in GUM. The mechanism captures long-tailed gradient updates distributed across layers and thereby enhances model memorization.

## 5.4 UNDERSTANDING THE EFFECT OF LAYERWISE SAMPLING

In this section, we investigate the underlying reason why the proposed algorithm of GUM can yield empirical improvements over GaLore. In short, GUM's high-rank gradient update leads to a more uniform singular value distribution in model parameters, which further results in more evenly

Table 4: **LLM Pre-training Results.** Trained models are evaluated on seven widely adopted commonsense reasoning tasks. All experiments are conducted on H100 GPUs.

| Model | Method | ARC-E | ARC-C | OBQA | HellaSwag | PIQA | SIQA | Winogrande | Avg. |
|-------|--------|-------|-------|------|-----------|------|------|------------|------|
| LLaMA-60M | FT-AdamW | 32.87 | 17.92 | 12.68 | 26.70 | 58.87 | 35.88 | 50.12 | 33.58 |
| | FT-Muon | 36.45 | 17.92 | 12.88 | 26.89 | 59.79 | 35.82 | 51.22 | 34.42 |
| | GaLore | 35.35 | 17.92 | 12.47 | 26.74 | 59.63 | 35.62 | 49.88 | 33.94 |
| | Fira | 35.02 | 18.94 | 12.27 | 26.75 | 58.71 | 36.24 | 50.28 | 34.03 |
| | Subtrack++ | 37.96 | 16.92 | 13.08 | 27.07 | 60.45 | 37.05 | 51.67 | **34.89** |
| | **GUM** | 36.28 | 17.41 | 13.68 | 26.70 | 60.12 | 36.54 | 51.85 | 34.65 |
| LLaMA-130M | FT-AdamW | 37.08 | 18.86 | 13.48 | 27.04 | 59.14 | 36.18 | 51.07 | 34.69 |
| | FT-Muon | 38.34 | 18.00 | 13.08 | 27.67 | 62.68 | 37.00 | 49.33 | 35.16 |
| | GaLore | 36.49 | 18.00 | 13.28 | 27.08 | 60.34 | 35.36 | 50.20 | 34.39 |
| | Fira | 26.01 | 19.54 | 12.27 | 26.13 | 53.65 | 34.19 | 49.80 | 31.66 |
| | Subtrack++ | 36.49 | 17.58 | 14.08 | 26.92 | 61.70 | 36.08 | 52.09 | 34.99 |
| | **GUM** | 38.01 | 18.34 | 14.69 | 27.32 | 61.26 | 36.44 | 52.49 | **35.51** |
| LLaMA-350M | FT-AdamW | 44.02 | 18.77 | 14.08 | 30.04 | 64.42 | 37.97 | 50.51 | 37.12 |
| | FT-Muon | 44.91 | 18.69 | 17.10 | 31.05 | 65.72 | 37.87 | 51.93 | 38.18 |
| | GaLore | 43.10 | 18.52 | 14.89 | 29.09 | 62.19 | 37.10 | 52.01 | 36.58 |
| | Fira | 42.38 | 18.77 | 15.49 | 29.27 | 63.00 | 37.97 | 51.85 | 36.96 |
| | Subtrack++ | 40.45 | 18.43 | 14.49 | 28.50 | 63.06 | 37.72 | 50.25 | 36.13 |
| | **GUM** | 44.44 | 19.80 | 15.69 | 29.28 | 64.53 | 38.13 | 51.38 | **37.42** |

distributed activations for input samples. This implies the long-tailed knowledge is better preserved in GUM-trained models, yielding better memorization.

**Setup.** We adopt the model of LLaMA-130M and benchmark of ARC-E (Clark et al., 2018), while keeping other settings the same as in Section 5.3.

**Results.** As shown in Figure 2, the overall stable ranks $\mathbb{E}\left[\|M\|_F^2/\|M\|_2^2\right]$ of GaLore and GUM are positively correlated with their performance in ARC-E, which provides direct evidence that higher stable ranks are generally beneficial for improving commonsense reasoning.

On top of that, it is observed in Figure 3 that GUM not only improves the overall stable rank of the trained model, but also shapes a set of more evenly distributed singular values in trained models, which further leads to more long-tail distributed activation across all modules. This provides indirect evidence and an intuitive explanation for the performance improvements in ARC-E: instead of overusing a low-dimensional space or a limited number of modules, GUM-trained models demonstrate a tendency to evenly distribute knowledge across all dimensions and modules, implying better memorization. Additional evidence is available in Appendix D.2.

## 6 CONCLUSIONS

In this paper, we investigate the debiasing technique of layerwise sampling for memory-efficient LLM training, whose combination with GaLore restores the theoretical convergence properties of full-parameter training. Our proposed algorithm, GUM, demonstrates that it is possible to achieve provable convergence in low-rank optimization without impairing its empirical performance and memory efficiency. Further analysis shows that the empirical gains are brought by the inherent high-rank updates, which lead to a higher overall stable rank and more uniformly distributed singular values in model parameters, yielding more long-tailed activation patterns and implying better memorization.

## ETHICS STATEMENT

After carefully reviewing the ethical regulations of the conference, to the best of our knowledge, this work does not present any foreseeable ethical concerns. No negative societal or ethical impacts are anticipated for the contribution of this work. The proposed algorithms are for general large language model training, and do not involve anything about human subjects, potentially harmful insights, potential conflicts of interest and sponsorship, discrimination/bias/fairness concerns, privacy and security issues, legal compliance, or research integrity issues.

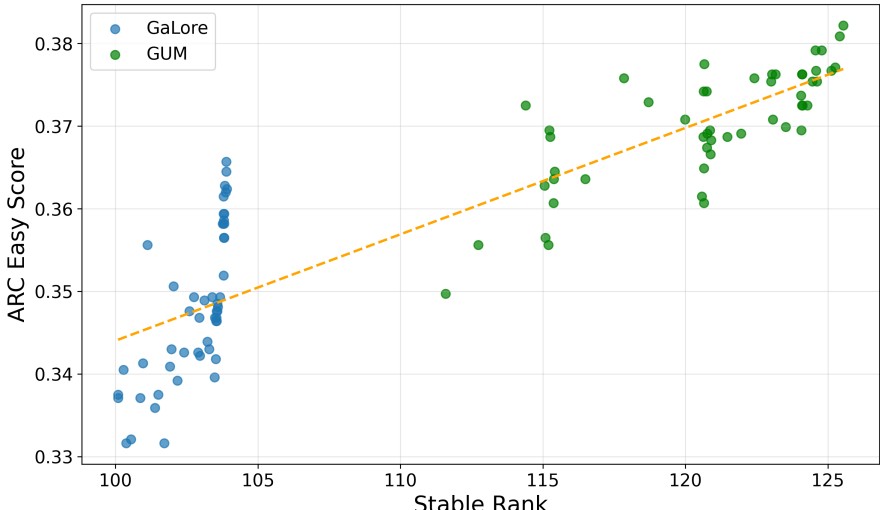

Figure 2: **Higher Stable Rank → Better Performance.** A positive correlation is observed between the overall stable rank $\mathbb{E}\left[\|M\|_F^2/\|M\|_2^2\right]$ and ARC Easy score. Each dot represents a checkpoint during pre-training after 1,000 steps, saved every 20 steps.

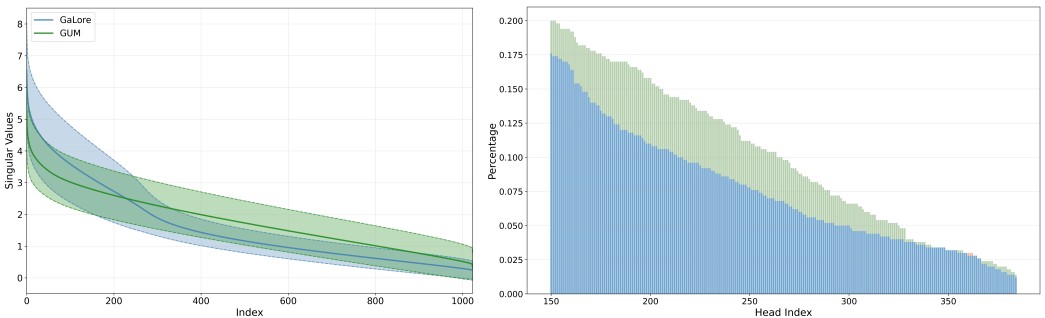

Figure 3: **Left: Updates → Weights**: Singular value distribution across layers of GaLore and GUM, where GUM demonstrates a more even and long-tailed distribution of singular values. **Right: Weights → Activations**: Tail distribution of modules that contain salient activations, where salient activations are defined as activations with top-k ($k = 10,000$) attention scores over all modules. Randomly sampled 1K inputs from the C4 corpus are utilized as prompts. Blue parts correspond to GaLore's tail distribution, while green parts stand for GUM's further increase on top of GaLore.

## REPRODUCIBILITY STATEMENT

We have made efforts to ensure that our work is reproducible, with details provided in Section 5 and Appendix C.

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

---

**Algorithm 3** An unbiased version of Algorithm 1

---

1: **Input:** $\{W_{0,\ell} \in \mathbb{R}^{m_\ell \times n_\ell}\}$ with each $\ell$ corresponding to the $\ell$-th block of parameters, number of blocks $N_L$, sampling period $K$, projection rank $r$
2: **for** $t = 0$ **to** $T/K - 1$ **do**
3:     `delete_optimizer_states()`     ▷ Delete the optimizer states to clear memory
4:     Each block $\ell$ is sampled to do full-rank updates with probability $q_{t,\ell}$
5:     **for** $k = 0$ **to** $K - 1$ **do**
6:       **for** $\ell = 1$ to $N_\ell$ **do**
7:         $G_{t,k,\ell} = G(W_{tK+k-1,\ell})$     ▷ Obtain the gradient of the $\ell$-th layer at $W_t$
8:         $P_{t,k,\ell} \leftarrow$ `get_projector()`     ▷ Obtain the projector $P_{t,k,\ell} \in \mathbb{R}^{m_\ell \times r}$
9:         $\tilde{G}_{t,k,\ell} = \begin{cases} \frac{1}{q_{t,\ell}}(I_{m_\ell} - P_{t,k,\ell}P_{t,k,\ell}^\top)G_{t,k,\ell}, & \text{if block } \ell \text{ is sampled to be full-rank} \\ \frac{1}{1-q_{t,\ell}}P_{t,k,\ell}^\top G_{t,k,\ell}, & \text{else} \end{cases}$
10:         $S_{t,k,\ell} \leftarrow$ `optimizer.update_state`$(\tilde{G}_{t,k,\ell})$     ▷ Run the base algorithm with $\tilde{G}_{t,k,\ell}$
11:         $W_{tK+k+1,\ell} = \begin{cases} W_{tK+k,\ell} - \eta S_{t,k,\ell}, & \text{if block } \ell \text{ is sampled to be full-rank} \\ W_{tK+k,\ell} - \eta P_{t,k,\ell}S_{t,k,\ell}, & \text{else} \end{cases}$
12:       **end for**
13:     **end for**
14: **end for**

---

# A    A GENERAL UNBIASED LOW-RANK GRADIENT METHOD PARADIGM

Here, we present our unbiased algorithm paradigm in Algorithm 3. The key idea of the algorithm is to compensate for biased errors introduced by the low-rank projection $P_t P_t^\top G_t$. To implement this while retaining memory efficiency, we refer to the main idea of LISA (Pan et al., 2024), which allows some of the blocks to be sampled uniformly with probability $q$ in each period. This compensated full-rank updates use $G_t - P_t P_t^\top G_t$, while other blocks still do the original low-rank update. By carefully balancing the scaling constants for the two different updates, the biased low-rank term can be canceled out in expectation, resulting in an unbiased estimation of gradients across iterations. This unbiased version of the algorithm is presented in Algorithm 3.

In one training process, the algorithm contains separated periods just like the vanilla GaLore algorithm (Zhao et al., 2024) and LISA (Pan et al., 2024). During each period $t$, each block of parameters is sampled to do full-rank updates with probability $q_{t,\ell}$. In each iteration $k$ in the period, we first compute the projection matrix $P_{t,k,\ell}$. Note that a lot of strategies for selecting projection matrices and sampling importance can be applied here (Guo et al., 2020; Sung et al., 2021; Ansell et al., 2021; Das et al., 2023; Muhamed et al., 2024; Ramesh et al., 2024; Liu et al., 2024a). Then, the blocks not sampled to do full-rank updates run basically the same low-rank update with Algorithm 1, while the full-rank blocks directly run the base optimizer with the compensated gradient $\tilde{G}_{t,k,\ell} = (I_m - P_{t,k,\ell}P_{t,k,\ell}^\top)G_{t,k,\ell}$.

We note that the proposed debiasing technique Algorithm 3 works generally when the following properties are satisfied:

- **Property I.** The columns of the low-rank projection matrix $P_t \in \mathbb{R}^{m \times r}$ with $r \leq m$ are orthonormal, i.e., $P_t^\top P_t = I_{r \times r}$.

- **Property II.** The projection and optimization updates are commutable, which means that $S_t = P_t$`optimizer.update_state`$(\tilde{G}_t) =$ `optimizer.update_state`$(P_t \tilde{G}_t)$. Optimizers satisfying this property typically treat the update parameters as matrices instead of vectors, and only conduct matrix operations in the update. Two standard examples include SGD and Muon (Jordan et al., 2024).

If the two properties are satisfied, we can show that Algorithm 3 is unbiased compared to the base optimizer, since it is equivalent to running the base optimizer with an unbiased estimation of the gradient at each iteration.

**Lemma 2** (Unbiased update of Algorithm 3). *When Property I and II are satisfied, a single iteration of Algorithm 3 for $W \in \mathbb{R}^{m \times n}$ is equivalent to*

$$S \leftarrow \texttt{optimizer.update\_state}(\hat{G})$$
$$W^+ = W - \eta S$$

*with $\mathbb{E}[\hat{G}] = G \in \mathbb{R}^{m \times n}$, where $G$ denotes the gradient obtained at $W$.*

## B    PROOFS OF SECTION 3 AND 4

### B.1    PROOF OF LEMMA 2 AND 1

*Proof of Lemma 2.* A single step of Algorithm 3 writes:

$$\tilde{G} = \begin{cases} \frac{1}{q}(I - PP^\top)G, & \text{with probability } q \\ \frac{1}{1-q}P^\top G, & \text{with probability } 1-q \end{cases}$$
$$S = \texttt{optimizer.update\_state}(\tilde{G})$$
$$W^+ = \begin{cases} W - \eta S, & \text{with probability } q \\ W - \eta PS, & \text{with probability } 1-q \end{cases}$$

where $G$ is the gradient at $W$ and $P$ is the projection matrix obtained at $W$. Based on the commutative property, we know that

$$W^+ = W - \eta PS = W - \eta \, \texttt{optimizer.update\_state}(P\tilde{G}),$$

which means that the update step is equivalent to

$$\hat{G} = \begin{cases} \frac{1}{q}(I - PP^\top)G, & \text{with probability } q \\ \frac{1}{1-q}PP^\top G, & \text{with probability } 1-q \end{cases}$$
$$S = \texttt{optimizer.update\_state}(\hat{G})$$
$$W^+ = W - \eta S$$

Since we have $\hat{G}$ is an unbiased estimation of $G$:

$$\mathbb{E}[\hat{G}] = q \cdot \frac{1}{q}(I - PP^\top)G + (1 - q) \cdot \frac{1}{1 - q}PP^\top G = G,$$

we finish the proof that Algorithm 3 is unbiased compared to the base optimizer. $\qquad\square$

*Proof of Lemma 1.* Based on Lemma 2, we only need to prove that GUM satisfies the two properties to finish the proof of Lemma 1.

**Property I.** Denote the projection matrix at one specific iteration $P$. Since $P$ is obtained from the SVD, we have $P \in \mathbb{R}^{m \times r}$ and $P^\top P = I_r$.

**Property II.** The base algorithm of GUM is Muon (Jordan et al., 2024). To prove the commutative property, we only need to prove that the Newton-Schulz iteration is commutable with $P$. In each iteration of the Newton Schulz iteration $\texttt{NewtonSchulz}(X)$, we compute

$$X^+ = aX + bXX^\top X + cXX^\top XX^\top X,$$

where $a, b, c \in \mathbb{R}$ are absolute constants. Then consider $\texttt{NewtonSchulz}(PX)$, we get

$$\begin{aligned} X^+ &= aPX + bPX(PX)^\top(PX) + cPX(PX)^\top(PX)(PX)^\top(PX) \\ &= P(aX + bXX^\top X + cXX^\top XX^\top X), \end{aligned}$$

where the second equality is because of Property I. Therefore, we obtain that

$$\texttt{NewtonSchulz}(PX) = P \cdot \texttt{NewtonSchulz}(X),$$

which finishes the proof of Property II and thus the unbiased property of GUM. $\qquad\square$

### B.2 PROOF OF THEOREM 1

We first state the notations in the following proof writing. For simplicity, we assume that the total iteration number $T = K\tau$. For $k = 0, \ldots, K-1$ in a specific period $t = 0, \ldots, \tau-1$, Algorithm 2 is mathematically equivalent to the following formulation:

$$\tilde{G}_{t,k} = \begin{cases} \frac{1}{1-q_t} P_t P_t^\top G_{t,k}, & \text{if } \xi_t = 0 \\ \frac{1}{q_t}(I - P_t P_t^\top) G_{t,k}, & \text{else} \end{cases}$$

$$\tilde{M}_{t,k} = \beta \tilde{M}_{t,k-1} + (1-\beta)\tilde{G}_{t,k}$$

$$W_{tK+k+1} = W_{tK+k} - \eta \texttt{NewtonSchulz}(\tilde{M}_{t,k})$$

where $G_{t,k}$ is the stochastic gradient obtained at $W_{tK+k}$ and $\xi_t \sim \text{Bernoulli}(q_t)$ is the indicator random variable such that $\xi_t = 1$ means using full-rank update in period $t$. We assume that the full-rank probability $q_t \equiv q$ and step size $\eta_t \equiv \eta$ are constants. The equivalence of Algorithm 2 and this formulation is shown by Lemma 1. At the beginning of each period, we initialize $P_t$ from $G_{t,0}$ and set $\tilde{M}_{t,-1} = 0$. Also, we denote $\nabla f_{t,k} \triangleq \nabla f(W_{tK+k})$ and $\text{msign}(X) \triangleq UV^\top$ for $X = U\Sigma V^\top$ as the SVD of $X$. Under Assumption 4, we have $\texttt{NewtonSchulz}(X) = \text{msign}(X)$. Note that here in the theoretical proof, we consider the damping, i.e., the $1 - \beta$ term in the update of $\tilde{M}_{t,k}$. Since we initialize $\tilde{M}_{t,k} = 0$ in each period, this damping will not affect the algorithm because the Newton-Schulz iteration is irrelevant to the input scale.

To help simplify the convergence proof, we also denote the residual of the projector as $R_t \in \mathbb{R}^{m \times (m-r)}$, i.e., we take $U_t = [P_t \ R_t] \in \mathbb{R}^{m \times m}$, which satisfies that $P_t^\top R_t = 0, R_t^\top P_t = 0$. Note that since we consider only the case $m \leq n$ here, we have $U_t U_t^\top = P_t P_t^\top + R_t R_t^\top = I$. We further define

$$Q_t \triangleq \begin{cases} P_t, & \text{if } \xi_t = 0 \\ R_t, & \text{else} \end{cases} \tag{3}$$

and the following auxiliary sequence

$$M_{t,k} = \beta M_{t,k-1} + (1-\beta)G_{t,k} \tag{4}$$

with $M_{t,-1} = 0$, which is the exponential moving average of the real gradient. With these definitions, we have

$$\text{msign}\left(\tilde{M}_{t,k}\right) = \text{msign}\left(Q_t Q_t^\top M_{t,k}\right) = Q_t \text{msign}\left(Q_t^\top M_{t,k}\right), \tag{5}$$

where the equation is based on the fact that $Q_t^\top Q_t = I$.

We first make use of the smoothness assumption to obtain a one-step analysis.

**Lemma 3** (One-step descent). *Under Assumption 2 and 4 and setting $\eta_t \equiv \eta$, for $t = 0, \ldots, \tau-1$ and $k = 0, \ldots, K-1$, it holds that*

$$f(W_{tK+k+1}) \leq f(W_{tK+k}) - \eta \left\| Q_t^\top \nabla f_{t,k} \right\|_* + \frac{1}{2}\eta^2 L_{\text{op}} + 2\eta \left\| M_{t,k} - \nabla f_{t,k} \right\|_*, \tag{6}$$

*where $Q_t$ is defined as (3).*

*Proof.* Based on Assumption 2, we have the descent property

$$f(W_{tK+k+1}) \leq f(W_{tK+k}) + \langle \nabla f_{t,K}, W_{tK+k+1} - W_{tK+k} \rangle + \frac{L_{\text{op}}}{2} \left\| W_{tK+k+1} - W_{tK+k} \right\|_{\text{op}}^2$$

$$= f(W_{tK+k}) - \eta \left\langle \nabla f_{t,K}, \text{msign}\left(\tilde{M}_{t,k}\right) \right\rangle + \frac{L_{\text{op}}\eta^2}{2} \left\| \text{msign}\left(\tilde{M}_{t,k}\right) \right\|_{\text{op}}^2$$

$$= f(W_{tK+k}) - \eta \left\langle M_{t,K}, \text{msign}\left(\tilde{M}_{t,k}\right) \right\rangle + \frac{L_{\text{op}}\eta^2}{2}$$

$$+ \eta \left\langle M_{t,K} - \nabla f_{t,k}, \text{msign}\left(\tilde{M}_{t,k}\right) \right\rangle$$

$$\leq f(W_{tK+k}) - \eta \left\langle M_{t,K}, \text{msign}\left(\tilde{M}_{t,k}\right) \right\rangle + \frac{L_{\text{op}}\eta^2}{2} + \eta \left\| M_{t,K} - \nabla f_{t,k} \right\|_*,$$

where the last inequality is based on the fact that $\|\cdot\|_*$ and $\|\cdot\|_{\mathrm{op}}$ are dual norms and $\left\|\mathrm{msign}\left(\tilde{M}_{t,k}\right)\right\|_{\mathrm{op}} = 1$. Then we further deal with the second term on the right hand side:

$$
\begin{aligned}
-\left\langle M_{t,K}, \mathrm{msign}\left(\tilde{M}_{t,k}\right)\right\rangle &\overset{(5)}{=} -\left\langle M_{t,K}, Q_t \mathrm{msign}\left(Q_t^\top M_{t,k}\right)\right\rangle \\
&= -\left\langle Q_t^\top M_{t,K}, \mathrm{msign}\left(Q_t^\top M_{t,k}\right)\right\rangle = -\left\|Q_t^\top M_{t,k}\right\|_* \\
&\leq -\left\|Q_t^\top \nabla f_{t,k}\right\|_* + \left\|Q_t^\top(M_{t,k} - \nabla f_{t,k})\right\|_* \\
&\leq -\left\|Q_t^\top \nabla f_{t,k}\right\|_* + \left\|M_{t,k} - \nabla f_{t,k}\right\|_*,
\end{aligned}
$$

where the last inequality is based on that $Q_t Q_t^\top \preceq I$. Then combining the inequalities, we can finish the proof. $\qquad\square$

Based on Lemma 3, we could find that a key to proving the convergence is the $\|M_{t,k} - \nabla f_{t,k}\|_*$ term. Let us define the following auxiliary sequences:

$$
\epsilon_{t,k} \triangleq M_{t,k} - \nabla f_{t,k}, \quad S_{t,k} \triangleq \nabla f_{t,k-1} - \nabla f_{t,k}, \quad N_{t,k} \triangleq G_{t,k} - \nabla f_{t,k} \tag{7}
$$

and additionally set $\nabla f_{t,-1} \triangleq \nabla f_{t,0}$ for all $t = 0, \ldots, \tau - 1$. Then we consider decomposing the desired $\epsilon_t$ based on the properties of moving average sequences.

**Lemma 4** (Decompose $\epsilon_{t,k}$). *For $t = 0, \ldots, \tau - 1$ and $k = 0, \ldots, K - 1$, it holds that*

$$
\epsilon_{t,k} = \sum_{i=1}^{k} \beta^{k-i+1} S_{t,i} + (1 - \beta) \sum_{i=0}^{k} \beta^{k-i} N_{t,i} - \beta^k \nabla f_{t,0}. \tag{8}
$$

*Proof.* From the definition of $M_{t,k}$ in (4), we know that

$$
M_{t,k} = \beta M_{t,k-1} + (1 - \beta) G_{t,k},
$$

which implies that

$$
\begin{aligned}
\epsilon_t &= \beta(M_{t,k-1} - \nabla f_{t,k-1}) + \beta(\nabla f_{t,k-1} - \nabla f_{t,k}) + (1 - \beta)(G_{t,k} - \nabla f_{t,k}) \\
&= \beta \epsilon_{t,k-1} + \beta S_{t,k} + (1 - \beta) N_{t,k}.
\end{aligned}
$$

Then by applying the equality recursively and noting that $\epsilon_{t,0} = (1 - \beta)G_{t,0} - \nabla f_{t,0} = (1 - \beta)N_{t,0} - \beta \nabla f_{t,0}$, we conclude the proof. $\qquad\square$

Then we produce the next lemma to state the variance contraction properties of momentum for Muon, which has been explored for Normalized SGD (Cutkosky & Mehta, 2020) and SignSGD (Sun et al., 2023), and also for Muon (Li & Hong, 2025), but with different assumptions.

**Lemma 5** (Variance Contraction). *Under Assumption 3, for $t = 0, \ldots, \tau - 1$ and $k = 0, \ldots, K - 1$, it holds that*

$$
\mathbb{E}\left[\left\|(1 - \beta) \sum_{i=0}^{k} \beta^{k-i} N_{t,i}\right\|_*\right] \leq \|V\|_* \sqrt{(1 - \beta^{2k})(1 - \beta)}. \tag{9}
$$

*Proof.* Based on Lemma 8 in An et al. (2025), for an arbitrary symmetric positive definite matrix $H \in \mathbb{R}^{m \times m}$, it holds that

$$
\begin{aligned}
\mathbb{E}\left[\left\|\sum_{i=0}^{k} \beta^{k-i} N_{t,i}\right\|_*\right] &\leq \mathbb{E}\left[\sqrt{\|H\|_* \mathrm{tr}\left(\left(\sum_{i=0}^{k} \beta^{k-i} N_{t,i}\right)^\top H^{-1} \left(\sum_{i=0}^{k} \beta^{k-i} N_{t,i}\right)\right)}\right] \\
&= \mathbb{E}\left[\sqrt{\|H\|_* \mathrm{tr}\left(\left(\sum_{i=0}^{k} \beta^{k-i} N_{t,i}\right) \left(\sum_{i=0}^{k} \beta^{k-i} N_{t,i}\right)^\top H^{-1}\right)}\right]
\end{aligned}
$$

$$\leq \sqrt{\|H\|_* \, \mathbb{E}\left[\mathrm{tr}\left(\left(\sum_{i=0}^{k}\beta^{k-i}N_{t,i}\right)\left(\sum_{i=0}^{k}\beta^{k-i}N_{t,i}\right)^{\top}H^{-1}\right)\right]}$$

$$= \sqrt{\|H\|_* \, \mathbb{E}\left[\mathrm{tr}\left(\left(\sum_{i=0}^{k}\beta^{2(k-i)}N_{t,i}N_{t,i}^{\top}\right)H^{-1}\right)\right]},$$

where the last inequality is based on the fact that $\mathbb{E}[\sqrt{X}] \leq \sqrt{\mathbb{E}[X]}$ and the last equality is based on the assumption that $N_{t,i}$ and $N_{t,j}$ are independent for $i \neq j$, which implies $\mathbb{E}[\mathrm{tr}\left(N_{t,i}N_{t,j}^{\top}H\right)] = 0$. Then taking $H = (VV^{\top})^{1/2}$ leads to

$$\sqrt{\|H\|_* \, \mathbb{E}\left[\mathrm{tr}\left(\left(\sum_{i=0}^{k}\beta^{2(k-i)}N_{t,i}N_{t,i}^{\top}\right)H\right)\right]} = \sqrt{\|V\|_* \, \mathbb{E}\left[\mathrm{tr}\left(\sum_{i=0}^{k}\beta^{2(k-i)}N_{t,i}N_{t,i}^{\top}(VV^{\top})^{-\frac{1}{2}}\right)\right]}$$

$$\leq \sqrt{\|V\|_* \sum_{i=0}^{k}\beta^{2(k-i)}\mathrm{tr}\left(VV^{\top}(VV^{\top})^{-\frac{1}{2}}\right)}$$

$$\leq \|V\|_* \sqrt{\frac{1-\beta^{2k}}{1-\beta^2}},$$

where the first inequality is based on Assumption 3 and the second inequality is by algebra. Then, combining the inequalities and multiplying $1 - \beta$ gives the result. $\square$

**Lemma 6** (Bound $\mathbb{E}\|\epsilon_{t,k}\|_*$). *Under Assumption 3, for $t = 0, \ldots, \tau - 1$ and $k = 0, \ldots, K - 1$, it holds that*

$$\mathbb{E}[\|\epsilon_{t,k}\|_*] \leq \frac{1-\beta^k}{1-\beta}L_{\mathrm{op}}\eta + \sqrt{(1-\beta^{2k})(1-\beta)}\,\|V\|_* + \beta^k\mathbb{E}\left[\|\nabla f_{t,0}\|_*\right]. \tag{10}$$

*Proof.* Based on Lemma 4, it holds that

$$\mathbb{E}\left[\|\epsilon_{t,k}\|_*\right] = \mathbb{E}\left[\left\|\sum_{i=1}^{k}\beta^{k-i+1}S_{t,i} + (1-\beta)\sum_{i=0}^{k}\beta^{k-i}N_{t,i} - \beta^k\nabla f_{t,0}\right\|_*\right]$$

$$\leq \sum_{i=1}^{k}\beta^{k-i+1}\|S_{t,i}\|_* + \left\|(1-\beta)\sum_{i=0}^{k}\beta^{k-i}N_{t,i}\right\|_* + \left\|\beta^k\nabla f_{t,0}\right\|_*,$$

where the inequality is based on the triangular inequality. For the first term in the RHS, it holds that

$$\|S_{t,i}\|_* = \|\nabla f_{t,i-1} - \nabla f_{t,i}\|_* \leq L_{\mathrm{op}}\|W_{tK+i-1} - W_{tK+i}\| = L_{\mathrm{op}}\eta.$$

Thus we have

$$\mathbb{E}\left[\|\epsilon_{t,k}\|_*\right] \leq \sum_{i=1}^{k}\beta^{k-i+1}L_{\mathrm{op}}\eta + \mathbb{E}\left[\left\|(1-\beta)\sum_{i=0}^{k}\beta^{k-i}N_{t,i}\right\|_*\right] + \mathbb{E}\left[\left\|\beta^k\nabla f_{t,0}\right\|_*\right]$$

$$\overset{(9)}{\leq} \sum_{i=1}^{k}\beta^{k-i+1}L_{\mathrm{op}}\eta + \sqrt{(1-\beta^{2k})(1-\beta)}\,\|V\|_* + \beta^k\mathbb{E}\left[\|\nabla f_{t,0}\|_*\right]$$

$$\leq \frac{1-\beta^k}{1-\beta}L_{\mathrm{op}}\eta + \sqrt{(1-\beta^{2k})(1-\beta)}\,\|V\|_* + \beta^k\mathbb{E}\left[\|\nabla f_{t,0}\|_*\right],$$

which concludes the proof. $\square$

We need to further determine the expected projected gradient for $\nabla f_{t,0}$.

**Lemma 7** (Expected projected gradient). *For $t = 0, \ldots, \tau - 1$, it holds that*

$$\mathbb{E}\left[\|Q_t^{\top}\nabla f_{t,0}\|_*\right] \geq \min\{q, 1-q\}\,\mathbb{E}\left[\|\nabla f_{t,0}\|_*\right]. \tag{11}$$

*Proof.* Based on the algorithm, we know that $\xi_t$ and $W_{tK}$ are independent, which means that

$$\mathbb{E}\left[\left\|Q_t^\top \nabla f_{t,0}\right\|_*\right] = (1-q)\mathbb{E}\left[\left\|P_t^\top \nabla f_{t,0}\right\|_*\right] + q\mathbb{E}\left[\left\|R_t^\top \nabla f_{t,0}\right\|_*\right].$$

Because we have $U_t = [P_t\ R_t]$ that satisfies $U_t^\top U_t = U_t U_t^\top = I$, it holds for any $X \in \mathbb{R}^{m\times n}$ that

$$\begin{aligned}
\left\|P_t^\top X\right\|_* + \left\|R_t^\top X\right\|_* &= \mathrm{tr}\left(\left(X^\top P_t P_t^\top X\right)^{\frac{1}{2}}\right) + \mathrm{tr}\left(\left(X^\top R_t R_t^\top X\right)^{\frac{1}{2}}\right) \\
&\geq \mathrm{tr}\left(\left(X^\top (P_t P_t^\top + R_t R_t^\top)X\right)^{\frac{1}{2}}\right) \\
&= \mathrm{tr}\left(\left(X^\top X\right)^{\frac{1}{2}}\right) = \|X\|_*.
\end{aligned}$$

Therefore, we have

$$\begin{aligned}
\mathbb{E}\left[\left\|Q_t^\top \nabla f_{t,0}\right\|_*\right] &= (1-q)\mathbb{E}\left[\left\|P_t^\top \nabla f_{t,0}\right\|_*\right] + q\mathbb{E}\left[\left\|R_t^\top \nabla f_{t,0}\right\|_*\right] \\
&\geq \min\{q, 1-q\}\left(\mathbb{E}\left[\left\|P_t^\top \nabla f_{t,0}\right\|_*\right] + \mathbb{E}\left[\left\|R_t^\top \nabla f_{t,0}\right\|_*\right]\right) \\
&\geq \min\{q, 1-q\}\mathbb{E}\left[\left\|\nabla f_{t,0}\right\|_*\right],
\end{aligned}$$

which completes the proof. $\qquad\square$

With the lemmas in hand, we are able to prove Theorem 1.

*Proof of Theorem 1.* Based on Lemma 3, for $t = 0, \ldots, \tau - 1$ and $k = 0, \ldots, K - 1$, it holds that

$$\begin{aligned}
f(W_{tK+k+1}) &\stackrel{(6)}{\leq} f(W_{tK+k}) - \eta\left\|Q_t^\top \nabla f_{t,k}\right\|_* + \frac{1}{2}\eta^2 L_{\mathrm{op}} + 2\eta\left\|M_{t,k} - \nabla f_{t,k}\right\|_* \\
&= f(W_{tK+k}) - \eta\left\|Q_t^\top \nabla f_{t,k}\right\|_* + \frac{1}{2}\eta^2 L_{\mathrm{op}} + 2\eta\left\|\epsilon_{t,k}\right\|_*,
\end{aligned}$$

where $Q_t$ is defined in (3) and $\epsilon_{t,k}$ is defined in (7). Then, after rearrangement and summation over $k$ and taking expectation, we have

$$\begin{aligned}
\sum_{k=0}^{K-1}\eta\mathbb{E}\left[\left\|Q_t^\top \nabla f_{t,k}\right\|_*\right] &\leq \mathbb{E}\left[f(W_{tK}) - f(W_{(t+1)K})\right] + \frac{1}{2}\eta^2 K L_{\mathrm{op}} + 2\eta\sum_{k=0}^{K-1}\mathbb{E}\left[\left\|\epsilon_{t,k}\right\|_*\right] \\
&\stackrel{(10)}{\leq} \mathbb{E}\left[f(W_{tK}) - f(W_{(t+1)K})\right] + \frac{1}{2}\eta^2 K L_{\mathrm{op}} \\
&\quad + 2\eta\sum_{k=0}^{K-1}\left(\frac{1-\beta^k}{1-\beta}L_{\mathrm{op}}\eta + \left(\sqrt{(1-\beta^{2k})(1-\beta)} + \beta^k\right)\|V\|_*\right) \\
&\leq \mathbb{E}\left[f(W_{tK}) - f(W_{(t+1)K})\right] + \eta^2 K L_{\mathrm{op}}\left(\frac{1}{2} + \frac{2(1-\beta^K)}{1-\beta}\right) \\
&\quad + 2\eta\left(\sqrt{(1-\beta^{2k})(1-\beta)}\|V\|_* + \beta^k\mathbb{E}\left[\left\|\nabla f_{t,0}\right\|_*\right]\right).
\end{aligned}$$

Since $W_{tK+k}$ is dependent on $Q_t$, it would be difficult to bound $\mathbb{E}[\|Q_t\nabla f_{t,k}\|_*]$ for $k \geq 1$. We therefore consider

$$\begin{aligned}
\sum_{k=0}^{K-1}\eta\mathbb{E}\left[\left\|Q_t^\top \nabla f_{t,k}\right\|_*\right] &\geq \sum_{k=0}^{K-1}\eta\mathbb{E}\left[\left\|Q_t^\top \nabla f_{t,0}\right\|_*\right] - \sum_{k=0}^{K-1}\eta\mathbb{E}\left[\left\|Q_t^\top (\nabla f_{t,k} - \nabla f_{t,0})\right\|_*\right] \\
&\geq \sum_{k=0}^{K-1}\eta\mathbb{E}\left[\left\|Q_t^\top \nabla f_{t,0}\right\|_*\right] - \sum_{k=1}^{K-1}\eta\mathbb{E}\left[\left\|\nabla f_{t,k} - \nabla f_{t,0}\right\|_*\right] \\
&\geq \sum_{k=0}^{K-1}\eta\mathbb{E}\left[\left\|Q_t^\top \nabla f_{t,0}\right\|_*\right] - \sum_{k=1}^{K-1}\eta\sum_{l=1}^{k}\mathbb{E}\left[\left\|\nabla f_{t,l} - \nabla f_{t,l-1}\right\|_*\right] \\
&\geq \sum_{k=0}^{K-1}\eta\mathbb{E}\left[\left\|Q_t^\top \nabla f_{t,0}\right\|_*\right] - \sum_{k=1}^{K-1}\eta L_{\mathrm{op}}\sum_{l=1}^{k}\mathbb{E}\left[\left\|W_{tK+l} - W_{tK+l-1}\right\|_{\mathrm{op}}\right]
\end{aligned}$$

$$\geq K\eta\mathbb{E}\left[\left\|Q_t^\top\nabla f_{t,0}\right\|_*\right] - \frac{K^2}{2}\eta^2 L_{\mathrm{op}},$$

where the first and third inequalities are based on the triangular inequality and the second inequality is based on that $Q_t Q_t^\top \preceq I$. The second last inequality uses Assumption 2. Then we combine the above inequalities and further sum up over $t$ and use Assumption 1 to obtain that

$$\sum_{t=0}^{\tau-1} K\mathbb{E}\left[\left\|Q_t^\top\nabla f_{t,0}\right\|_*\right] \leq \frac{f(W_0)-f^*}{\eta} + \eta K\tau L_{\mathrm{op}}\left(\frac{K+1}{2} + \frac{2(1-\beta^K)}{1-\beta}\right)$$
$$+ 2\tau K\sqrt{(1-\beta^{2K})(1-\beta)}\left\|V\right\|_* + \sum_{t=0}^{\tau-1}\frac{2(1-\beta^K)}{1-\beta}\mathbb{E}\left[\left\|\nabla f_{t,0}\right\|_*\right].$$

Combining Lemma 7, we have

$$K\mathbb{E}\left[\left\|Q_t^\top\nabla f_{t,0}\right\|_*\right] - \frac{2(1-\beta^K)}{1-\beta}\mathbb{E}\left[\left\|\nabla f_{t,0}\right\|_*\right] \geq \frac{K\alpha}{2}\mathbb{E}\left[\left\|\nabla f_{t,0}\right\|_*\right]$$

where $\alpha \triangleq \min\{q, 1-q\}$ and we take $\alpha > \frac{2}{K}$ and $1-\beta \geq \frac{2}{K\alpha}$. Thus, we can obtain that

$$\frac{\alpha}{2\tau}\sum_{t=0}^{\tau-1}\mathbb{E}\left[\left\|\nabla f_{t,0}\right\|_*\right] \leq \frac{f(W_0)-f^*}{\eta T} + \eta L_{\mathrm{op}}\left(\frac{K+1}{2} + \frac{2}{1-\beta}\right) + 2\sqrt{1-\beta}\left\|V\right\|_*$$
$$\leq \frac{f(W_0)-f^*}{\eta T} + \eta L_{\mathrm{op}}\left(\frac{K+1}{2} + K\alpha\right) + 2\sqrt{1-\beta}\left\|V\right\|_*$$

By choosing the hyperparameter as

$$\eta = \sqrt{\frac{TL_{\mathrm{op}}\left(\frac{K+1}{2} + K\alpha\right)}{f(W_0)-f^*}}, \quad \beta = 1 - \frac{2}{K\alpha}, \quad K = \max\left\{1, \min\left\{\frac{\sigma\sqrt{T}}{\sqrt{\alpha L(f(W_0)-f^*)}}, T\right\}\right\}, \tag{12}$$

we can obtain that

$$\min_{t=0,\ldots,\frac{T}{K}-1}\mathbb{E}\left[\left\|\nabla f(W_{tK})\right\|\right] \leq \mathcal{O}\left(\frac{1}{\alpha}\sqrt{\frac{L_{\mathrm{op}}\Delta}{T}} + \left(\frac{L_{\mathrm{op}}\Delta\left\|V\right\|_*^2}{\alpha^5 T}\right)^{\frac{1}{4}} + \frac{\left\|V\right\|_*}{\sqrt{\alpha^3 T}}\right),$$

with $\Delta \triangleq f(W_0) - f^*$, which finishes the proof. □

## C  TRAINING SETUP AND HYPERPARAMETERS

### C.1  FINE-TUNING SETUP

In our experiments, we slightly modify the full-rank update rule (2) for GUM by multiplying $(1-q_{t,\ell})$ on $-P_{t,\ell}P_{t,\ell}^\top G_{t,k,\ell}$. This modification still preserves the unbiased property while being able to recover the original full-parameter Muon algorithm by setting $q_{t,\ell} = 1$.

We utilize LMFlow (Diao et al., 2023)[2] to perform full-parameter fine-tuning, GaLore tuning, and GUM tuning. We set the number of training epochs for all fine-tuning scenarios to 1. All experiments were conducted on a single NVIDIA H100 GPU with 80 GB of memory.

We explored a range of learning rates from $8\times 10^{-6}$ to $1\times 10^{-4}$, applying this range to Full Parameter Training, GaLore, and GUM. For GaLore, we fixed the rank $r = 512$ and applied it uniformly across all layers. In the case of GUM, the number of layers ($\gamma$) selected for full-rank updates was set to 2 for all models. The sampling interval $K$, which defines the number of update steps between each layer selection, was varied between 10 and 300, depending on factors such as dataset size, batch size, and total training steps. The models covered in this paper can be found in Table 5.

---

[2]https://github.com/OptimalScale/LMFlow

Table 5: Baseline Model Configurations

| Model Name | # Params | # Layers | Model Dim |
|------------|----------|----------|-----------|
| LLAMA-3-8B | 8 B | 32 | 4096 |
| QWEN-2.5-7B | 7 B | 28 | 3584 |
| GEMMA-2-9B | 9 B | 42 | 3584 |

## C.2 FINE-TUNING HYPERPARAMETERS

We began our study by conducting a grid search over two key hyperparameters: (i) the learning rate and (ii) the number of sampling layers used for full-rank updates. Given the strong empirical performance of the GaLore method, we fixed the rank to $r = 512$. The learning rate was explored within the range $\{8 \times 10^{-6}, 2 \times 10^{-5}, 4 \times 10^{-5}, 6 \times 10^{-5}, 8 \times 10^{-5}, 1 \times 10^{-4}\}$, applied consistently across full parameter training, GaLore, and GUM. For GaLore, we followed the official `Transformers` implementation [3], using the default settings and aligning the learning rate with the full parameter training. With respect to the number of sampling layers, and in accordance with Table 3, we selected values that did not exceed the GPU memory cost of GaLore. As a result, $\gamma = 2$ was used in most GUM configurations. The sampling period $K$ was uniformly set to 200 for all models. A detailed summary of the optimal hyperparameter values identified for each setting is provided in Table 6.

Table 6: Optimal settings for each method were determined through hyperparameter search: FT (Full-parameter Training)-AdamW, FT-Muon, GaLore, and GUM.

| Model | FT-AdamW | FT-Muon | GaLore | | GUM | | |
|-------|----------|---------|--------|------|-----|---|---|
| | lr | lr | lr | Rank | lr | $\gamma$ | $K$ |
| LLaMA-3-8B | $3 \times 10^{-5}$ | $7 \times 10^{-5}$ | $9 \times 10^{-5}$ | 512 | $1 \times 10^{-4}$ | 2 | 200 |
| Qwen-2.5-7B | $1 \times 10^{-5}$ | $5 \times 10^{-5}$ | $7 \times 10^{-5}$ | 512 | $7 \times 10^{-5}$ | 2 | 200 |
| Gemma-2-9B | − | $4 \times 10^{-5}$ | $4 \times 10^{-5}$ | 512 | $6 \times 10^{-5}$ | 2 | 200 |

## C.3 PRE-TRAINING HYPERPARAMETERS

In our experiments, we utilize C-optim [4] for the pre-training. Following standard protocol, we fixed the LLaMA context length to 1024 tokens. Similar to the fine-tuning setup, we made a grid search on learning rate and the number of sampling layers. The sampling period $K$ was set to 100 for 130M and 350M models, 50 for the 60M model. A detailed summary of the optimal hyperparameter values identified for each setting is provided in Table 7.

Table 7: Optimal settings for each method were determined through hyperparameter search: AdamW, Muon, Fira, GaLore, and GUM.

| Model | AdamW | Muon | Fira | | GaLore | | GUM | | |
|-------|-------|------|------|------|--------|------|-----|---|---|
| | lr | lr | lr | Rank | lr | Rank | lr | $\gamma$ | $K$ |
| LLaMA-60M | $3 \times 10^{-3}$ | $1 \times 10^{-2}$ | $9 \times 10^{-3}$ | 256 | $9 \times 10^{-3}$ | 256 | $9 \times 10^{-3}$ | 4 | 50 |
| LLaMA-130M | $2 \times 10^{-3}$ | $5 \times 10^{-3}$ | $5 \times 10^{-3}$ | 256 | $5 \times 10^{-3}$ | 256 | $5 \times 10^{-3}$ | 4 | 100 |
| LLaMA-350M | $1 \times 10^{-3}$ | $3 \times 10^{-3}$ | $3 \times 10^{-3}$ | 256 | $3 \times 10^{-3}$ | 256 | $3 \times 10^{-3}$ | 6 | 100 |

---

[3]https://github.com/jiaweizzhao/GaLore
[4]https://github.com/kyleliang919/C-Optim

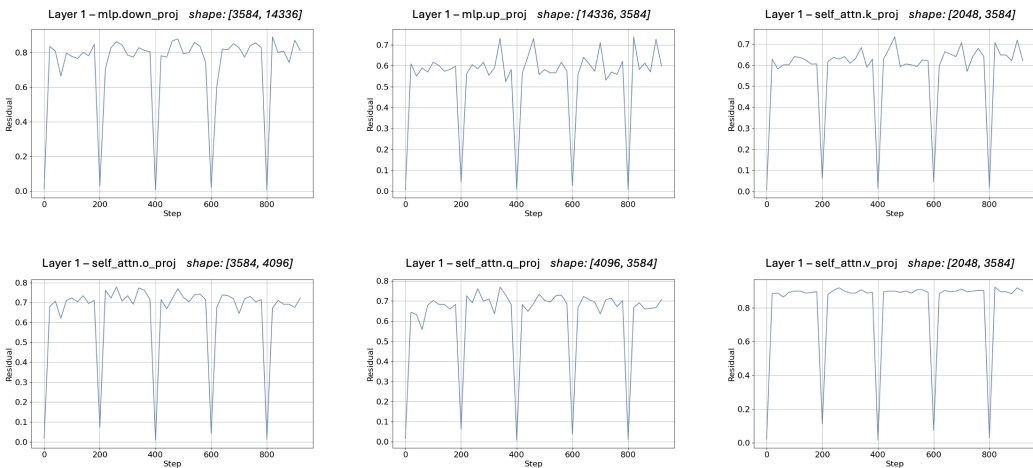

Figure 4: Residual ($\chi_t = \|G_t^u - G_t^p\|_{\mathrm{F}}/\|G_t^u\|_{\mathrm{F}}$) between GaLore's projected and original gradients across different blocks during **Gemma-2-9B** fine-tuning. High residuals persist throughout training (except for the iterations with projector updates), revealing systematic bias in GaLore updates.

## D  ADDITIONAL EXPERIMENTAL RESULTS

### D.1  BIAS IN GALORE

To further illustrate how significant the bias in low-rank projection methods is, we analyze the residuals between low-rank projected gradients and the original full-rank gradients across multiple layers during the fine-tuning of the **Gemma-2-9B** model on the **GPT-4-LLM** dataset. The residual is computed as follows:

$$\chi_t = \frac{\|G_t^u - G_t^p\|_{\mathrm{F}}}{\|G_t^u\|_{\mathrm{F}}}, \tag{13}$$

where $G_t^u$ represents the original gradient at iteration $t$ without projection, and $G_t^p$ denotes the low-rank projected gradients in GaLore-Muon. We can see that $\chi_t$ presents the relative error between the original gradients and the projected gradients at iteration $t$, showing how much the projection operation makes the gradient estimation biased from the original one. We measure this relative error for each block of parameters along the trajectory of the GaLore-Muon algorithm every 20 iterations. The projector update frequency is set to 200, and the projection rank is 512. We use a batch size of 16 and a learning rate of $7 \times 10^{-5}$. For demonstration purposes, we specifically select the self-attention and MLP weights at layer 10.

As depicted in Figure 4, the relative error shows a periodic curve. It is relatively small (around $0 - 20\%$) in the iteration $t$ such that $t$ is a multiple of the update frequency 200, where the projector is updated based on the gradient. Since the GaLore projector is chosen as the singular vectors of the largest singular values of the current gradient, it is a good low-rank projector for the current gradient, which results in this small error. [5] However, we can see that the relative error rapidly increases after this and achieves even higher than $60 - 80\%$ in less than 20 iterations. This implies that although the low-rank projection of GaLore doesn't hurt much in the first iteration, it makes little sense for the following gradients, since the projection produces a really high relative error. Such a high relative error demonstrates a remarkably significant bias between the low-rank projected gradients and the original gradients, and between GaLore and the original gradient algorithm, highlighting the need to derive an unbiased low-rank projection algorithm.

---

[5]Note that while the projector is good for the stochastic gradient used in the algorithm, it can still be a large obstacle to the convergence, as shown in Figure 1.

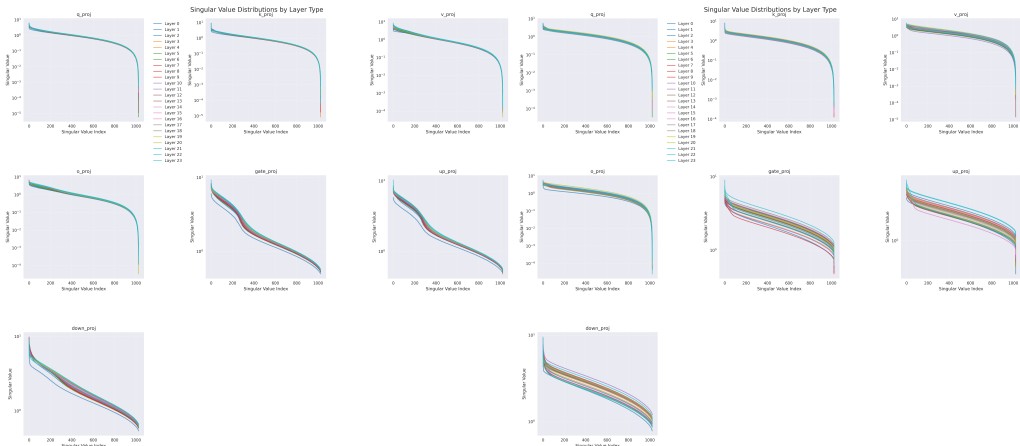

Figure 5: **Detailed Singular Value Distribution. Left**: GaLore. **Right**: GUM. It can be observed that GaLore has a sudden magnitude drop in the tail distribution of singular values in `gate proj` and `up proj` modules. GUM generally demonstrates smoother and more long-tailed singular value distributions. Furthermore, GUM has a differentiated spectrum across different layers, while this phenomenon is much weaker in GaLore.

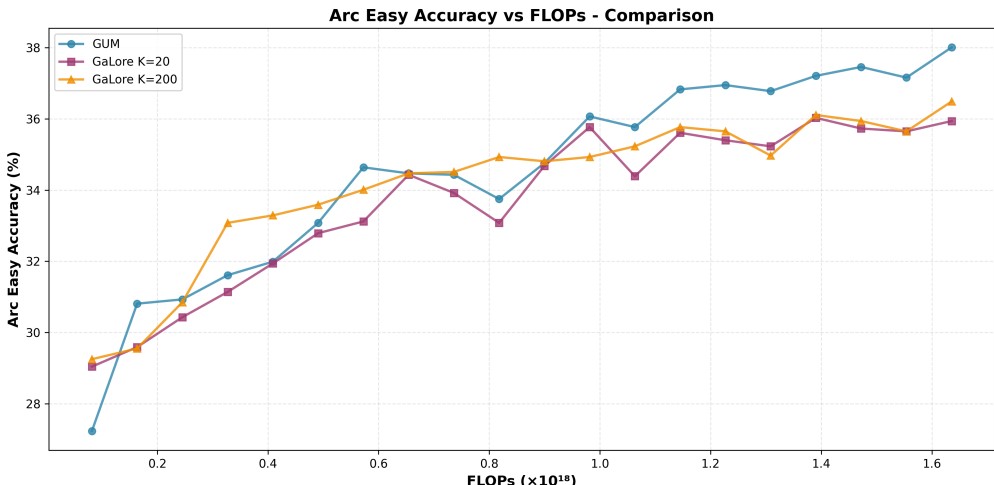

Figure 6: **Computational Cost Comparison.** The quality-vs.-time curve of GaLore with $K = 20/200$ projector refreshing period and GUM.

### D.2 SINGULAR VALUE DISTRIBUTION OF MODEL WEIGHTS

As shown in Figure 5, GUM demonstrates a smoother and more long-tailed singular value distribution than GaLore, especially in modules of `gate proj` and `up proj`. The spectrums are also more differentiated and have a non-trivial diversity across layers in GUM.

### D.3 COMPUTATIONAL COST COMPARISON

To compare the computational cost between GaLore and GUM, additional experiments on LLaMA-130M are conducted, following the same setting in Section 5.3. Results of different projector refreshing periods $K = 20/200$ for GaLore are also included to understand the effect of projector staleness. As shown in Figure 6, GUM is more computationally efficient than GaLore, and the projector refreshing period has little effect on GaLore's computational efficiency.

Table 8: **Ablation studies** for different choice of #sampled full-rank layers, sampling period $K$, and rank $r$. The default setup is #sampled full-rank layers = 4, $K$=200, and $r$=128.

| #Sampled full-rank layers | Prompt-level Strict Accuracy | Prompt-level Loose Accuracy | $K$ | Prompt-level Strict Accuracy | Prompt-level Loose Accuracy | $r$ | Prompt-level Strict Accuracy | Prompt-level Loose Accuracy |
|---|---|---|---|---|---|---|---|---|
| 1 | 30.87 | 34.20 | 20 | 29.21 | 33.83 | 32 | 30.98 | **36.89** |
| 2 | 31.61 | 35.12 | 100 | 30.68 | 35.21 | 64 | 31.24 | 36.60 |
| 4 | 33.27 | 36.60 | 200 | **33.27** | **36.60** | 128 | **33.27** | 36.60 |
| 6 | **33.39** | **36.75** | 500 | 29.39 | 33.27 | | | |
| 10 | 32.36 | 36.16 | | | | | | |
| 20 | 28.36 | 30.76 | | | | | | |

Table 9: **LLM Fine-tuning with More Baselines.** Trained models are evaluated on IFEval (instruction-following) and GSM8K (mathematical reasoning) with Qwen-2.5-7B model. All experiments are conducted on a single H100 GPU.

| Method | IFEval | | GSM8K |
|---|---|---|---|
| | Prompt-level Strict-Accuracy | Prompt-level Loose-Accuracy | Accuracy |
| LDAdamW (Robert et al., 2025) | 28.10 | 30.31 | 83.78 |
| Apollo (Zhu et al., 2025) | 31.61 | 36.41 | 85.67 |
| Subtrack++ (Rajabi et al., 2025) | 29.76 | 34.01 | 86.66 |
| GUM | **33.46** | **38.82** | **86.81** |

## D.4 ABLATION STUDIES

To better understand the tradeoff between sampling probability, sampling period, and ranks, additional ablation studies are conducted following the setting of Section 5.2. All experiments are conducted on IFEval benchmark with Gemma-2-9B.

As shown in the table above, the best choice of sampled layers is 6, where the performance starts to degrade when more full-rank layers are introduced. This is consistent with the observation in Pan et al. (2024), where this sampling-style training is conjectured to introduce an implicit regularization effect for supervised fine-tuning tasks.

For the best sampling period $K$, our choice of $K = 200$ is already optimal, where a smaller $K$ may compromise the momentum and decelerate the training process, while a larger $K$ results in insufficient sampling of all layers.

For the best rank $r$, increasing the rank from 32 to 128 leads to overall performance improvements, especially in prompt-level strict accuracy. This means the higher-rank update captures more details for following the given instruction.

## D.5 COMPARISON WITH MORE BASELINES

To further highlight GUM's performance, we have included comparisons with LDAdamW (Robert et al., 2025), Apollo (Zhu et al., 2025), and SubTrack++ (Rajabi et al., 2025) on IFEval and GSM8K using the Qwen-2.5-7B model.

## D.6 LARGER-SCALE PRE-TRAINING

To verify GUM's effectiveness in pre-training on larger-sized model tasks, we conducted an additional pre-training experiment on a 7B-sized LLaMA model. Due to computational resource constraints, we follow the setup in SubTrack++ (Rajabi et al., 2025), and report results using the same number of tokens for pre-training.

As shown in the table, GUM outperforms GaLore, AdamW, and Fira in 7B-sized models as well, with better or no-worse performance on 4 out of 7 tasks, and a higher overall accuracy.

Table 10: **LLM Pre-training with 7B-sized LLaMA.** Trained models are evaluated on seven widely adopted commonsense reasoning tasks. All experiments are conducted on H100 GPUs.

| Method | ARC-E | ARC-C | OBQA | HellaSwag | PIQA | SIQA | Winogrande | Avg. |
|--------|-------|-------|------|-----------|------|------|------------|------|
| GaLore | 26.39 | 19.80 | 12.27 | 25.85 | 52.77 | 34.60 | 49.88 | 31.65 |
| AdamW | 26.68 | 18.94 | 13.48 | 26.11 | 51.69 | 34.54 | 50.70 | 31.73 |
| Fira | 26.46 | 20.90 | 13.01 | 25.65 | 52.34 | 34.75 | 50.54 | 31.95 |
| **GUM** | 26.64 | 20.90 | 12.27 | 25.83 | 53.75 | 35.26 | 50.91 | **32.22** |

## D.7 LONGER TRAINING

To verify GUM's effectiveness in longer training scenarios, we extended the number of epochs and conducted additional experiments on the IFEval benchmark with Gemma-2-9B.

Table 11: **LLM Fine-tuning with Longer Training.** Trained models are evaluated on IFEval (instruction-following) with Gemma-2-9B model. All experiments are conducted on a single H100 GPU.

| #Epoch | IFEval | |
|--------|--------|--------|
| | Prompt-level Strict-Accuracy | Prompt-level Loose-Accuracy |
| 1 | 33.27 | 36.60 |
| 3 | 29.57 | 31.42 |
| 5 | 26.25 | 28.28 |

As shown in Table 11, the performance degrades with an increasing number of epochs, indicating overfitting. So the number of epochs is sufficient for this supervised fine-tuning setting.

To further investigate GUM's performance in effectively longer training settings, we conducted additional experiments on LLaMA-60M in Table 4, increasing the data amount to 5B tokens (originally 1.5B).

As shown in Table 12, GUM still outperforms Fira and GaLore by a non-trivial margin, demonstrating the effectiveness of GUM under longer training settings.

## E FURTHER DISCUSSION

### E.1 RELATIONSHIP WITH MUON OPTIMIZER

It is worth noticing that main focus of GUM is not the Muon optimizer (Jordan et al., 2024), but a technique for debiasing existing low-rank training methods like GaLore (Zhao et al., 2024), which is empirically orthogonal to the underlying optimizers such as AdamW and Muon.

Regarding Muon's properties, there are several points worth mentioning:

- There is a fundamental tradeoff between Muon and AdamW across different model sizes.
  - Generally, Muon favors *deep and thin* networks, while AdamW has memory advantages in *large-scale wide* networks. On one hand, Muon may incur higher memory cost for extremely large hidden layers, since Muon requires matrix–matrix multiplication in the Newton–Schulz5 update, whereas AdamW only requires matrix–vector multiplication operations. On the other hand, Muon has only one momentum term, while AdamW has an additional second moment, which incurs extra memory consumption.
  - For commonly used ~7B-sized models like Gemma-2-9B, AdamW empirically requires more GPU memory, as shown in Table 2, where AdamW triggers an out-of-memory error while Muon does not.

Table 12: **LLM Pre-training with LLaMA-60M with long training (1.5B → 5B tokens).** Trained models are evaluated on seven widely adopted commonsense reasoning tasks. All experiments are conducted on H100 GPUs.

| Method | ARC-E | ARC-C | OBQA | HellaSwag | PIQA | SIQA | Winogrande | Avg. |
|--------|-------|-------|------|-----------|------|------|------------|------|
| GaLore | 37.50 | 18.60 | 11.07 | 27.07 | 60.66 | 37.56 | 49.57 | 34.58 |
| Fira | 37.16 | 17.66 | 14.29 | 27.38 | 60.72 | 37.62 | 50.83 | 35.09 |
| **GUM** | 37.79 | 17.75 | 14.29 | 27.30 | 61.26 | 37.81 | 51.67 | **35.41** |

- Muon has been successfully applied to Mixture-of-Experts training (Kimi, 2025) and outperforms AdamW, as shown in (Liu et al., 2025a).
- **Why do we choose Muon as the base optimizer in GUM?** Muon performs well in large LLMs, as demonstrated by Kimi K2 (Kimi, 2025), which has 1T total parameters and 32B activated parameters. On the empirical side, Muon is demonstrated to perform better than AdamW in pre-training tasks (Liu et al., 2025a; Kimi, 2025). In addition, Muon incurs less memory consumption for common ∼7B-sized models since it has no second moments. On the theoretical side, Muon satisfies properties I and II in Lemma 2, allowing the unbiasedness to be proven.

## F    BROADER IMPACTS

Memory-efficient training techniques are critical for scalable LLM development and for democratizing customized LLMs for broader societal use. Improving theoretical guarantees provides insights for the invention of new methods with enhanced performance, leading to reduced computational resource consumption and lower carbon dioxide emissions.

## G    LIMITATIONS

The technique of sampled high-rank updates inherently introduces high variance into the per-iteration updates when the sampling probability is low, which leads to instability in the training procedure and requires more careful tuning of the hyperparameters. To alleviate this issue, standard theoretical tools for variance reduction can be employed (Johnson & Zhang, 2013; Needell et al., 2014; Ge et al., 2019b), which we leave for future work here. The analysis can also be combined with other acceleration (Zhang & Xiao, 2017; Ge et al., 2019a; Pan et al., 2021; 2023; Defazio et al., 2024; Liu et al., 2025b) and generalization techniques (Arjovsky et al., 2019; Foret et al., 2020; Hao et al., 2025), whose properties are worth investigating as open problems. The algorithm's empirical performance and computational cost in other types of models (Devlin et al., 2019; Rombach et al., 2022; Pan et al., 2022; Liu et al., 2023; Gu & Dao, 2023; Hu et al., 2024; Wang et al., 2025; Mu & Lin, 2025) and applications (Xia et al., 2023; Peebles & Xie, 2023; Pan et al., 2025) also remain as interesting questions.

## H    THE USE OF LARGE LANGUAGE MODELS

ChatGPT and GPT-5 were adopted to polish the writing of the paper, where all revised sentences were double-checked by the authors. OpenAI Deep Research was utilized for finding dataset licenses.

## I    LICENSES

For mathematical reasoning tasks in LLM fine-tuning, the training dataset comes from 4 different sources: DART-Math (Tong et al., 2024), UltraInteract (Yuan et al., 2024), MathInstruct (Yue et al., 2023), and Orca-Math (Mitra et al., 2024), with their licenses listed in Table 13. Other datasets and benchmarks are also available in the same table.

| Training Datasets | #Samples | Kind | License |
|---|---|---|---|
| `teknium/GPT4-LLM-Cleaned`[6] | 55K | Instruction | CC BY-NC 4.0 |
| DART-Math[7] (Tong et al., 2024) | 591K | Math | MIT |
| `openbmb/UltraInteract_sft`[8] (Yuan et al., 2024) | 289K | Reasoning | MIT |
| `TIGER-Lab/MathInstruct`[9] (Yue et al., 2023) | 262K | Reasoning | MIT |
| `microsoft/orca-math-word-problems-200k`[10] (Mitra et al., 2024) | 200K | Math | MIT |
| C4 corpus[11] (Raffel et al., 2023) | >1B | Commonsense | ODC-BY |
| IFEval (Zhou et al., 2023) | 0.5K | Instruction | Apache-2.0 |
| GSM8K (Cobbe et al., 2021b) | 7.5K | Math | MIT |
| ARC-E (Clark et al., 2018) | 5.2K | Instruction | CC-BY-SA-4.0 |
| ARC-C (Clark et al., 2018) | 2.6K | Instruction | CC-BY-SA-4.0 |
| HellaSwag (Zellers et al., 2019b) | 10K | Commonsense reasoning | MIT |
| PIQA[12] (Bisk et al., 2020) | 3K | Commonsense reasoning | Academic Free License v. 3.0 |
| SIQA (Sap et al., 2019) | 2.2K | Commonsense reasoning | CC-BY-4.0 (Li et al., 2024) |
| Winogrande[13] (Sakaguchi et al., 2021b) | 1.8K | Commonsense reasoning | CC-BY |
| OBQA (Mihaylov et al., 2018a) | 5.9K | Commonsense reasoning | (permissive open license)[14] |

Table 13: Licenses of training datasets and benchmarks. Here, the number of samples for benchmarks only counts the test set.

For code repositories, LMFlow (Diao et al., 2023) is released under Apache-2.0 license.

---

[6] https://huggingface.co/datasets/teknium/GPT4-LLM-Cleaned
[7] https://huggingface.co/datasets/hkust-nlp/dart-math-uniform
[8] https://huggingface.co/datasets/openbmb/UltraInteract_sft
[9] https://huggingface.co/datasets/TIGER-Lab/MathInstruct
[10] https://huggingface.co/datasets/microsoft/orca-math-word-problems-200k
[11] https://huggingface.co/datasets/allenai/c4
[12] https://github.com/ybisk/ybisk.github.io/tree/master/piqa
[13] https://github.com/allenai/winogrande
[14] The OpenBookQA dataset is released under a permissive open license, making it freely available for academic research. In practice, sources indicate that the dataset is in the public domain or under a very permissive license. For example, a Kaggle distribution of OpenBookQA explicitly labels it CC0 1.0 Universal (Public Domain) (https://www.kaggle.com/datasets/thedevastator/openbookqa-a-new-dataset-for-advanced-question-a). Similarly, a curated dataset list reports OpenBookQA's license as Apache 2.0 (https://github.com/lmmlzn/Awesome-LLMs-Datasets). Both of these licenses allow unrestricted use, redistribution, and modification of the data, including for academic purposes.

