# OpenReview forum: "Unbiased Gradient Low-Rank Projection"
_ICLR.cc/2026/Conference — Submitted to ICLR 2026_

### Official Review · Reviewer_SdfL · 2025-10-28

**Soundness:** 3
**Presentation:** 3
**Contribution:** 2
**Rating:** 6
**Confidence:** 3

**Summary:**

This paper proposes GUM, a new low-rank optimization algorithm aiming to achieve unbiased gradient updates for memory-efficient LLM training. GUM combines the low-rank gradient projection mechanism of GaLore with the Muon optimizer, augmented by a layerwise-sampling debiasing technique. The key idea is to probabilistically perform full-rank updates on a subset of layers, ensuring that the overall gradient estimation remains unbiased in expectation.

**Strengths:**

- The paper introduces a new approach to mitigating the bias introduced by low-rank projections through layerwise sampling. This idea extends beyond GaLore and establishes a general theoretical analysis for constructing unbiased low-rank optimizers.

 - The authors provide a formal convergence theorem (Theorem 1) under non-convex settings, showing that GUM retains the same convergence rate as Muon. The theoretical derivations are logical and clear.

**Weaknesses:**

- W1. Unbiased gradient statement as key contribution is overstated. This property requires taking the expectation over stochastic sampling. Moreover, as the full gradient evolves along the training trajectory, it is unclear to which the step of the full gradient the unbiasedness claim refers. Therefore, the claim of unbiasedness appears to be overstated and lacks rigorous justification.

 - W2. According to Theorem 1, the optimal choice of q  is 0.5, which coincides with the hyperparameter used in the experiments (Line 290). However, using a constant q implies that the full-rank updates dominate the space complexity, matching that of full fine-tuning. This significantly limits the contribution of the work. It would be more meaningful to design an algorithm with a diminishing q, aligning better with the paper’s goal of memory-efficient optimization.

 - W3. In Section 5.1, the authors do not use q = 0.5; instead, they only update two layers with full-rank gradients. This further indicates that the proposed method is not an unbiased gradient estimator for low-rank projection under efficient memory constraints, undermining the claimed theoretical properties.

 - W4. Insufficient training epochs for convergence. For the 8B-parameter model with rank = 512, the reported number of epochs appears insufficient for convergence. The authors are encouraged to extend training to at least 5 epochs to provide a more convincing empirical evaluation.

**Questions:**

- Since the authors position GoLore as the closest approach to building an unbiased algorithm, the absence of direct empirical comparison makes the conclusions about landscape properties and convergence rates unconvincing.
 - What is the key innovation in using the optimizer Muon for low-rank adaptation during fine-tuning in this work? The paper does not clearly state why this optimizer provides unique benefits compared to standard choices (e.g., AdamW).

---

> ### Author Response · Authors · 2025-11-21
> **R1: Response to Reviewer SdfL - Part 1**
>
> Thank you so much for your valuable suggestions and insightful comments. We deeply appreciate your recognition of the novelty and theoretical contribution of our work. We would like to address your concerns as follows:
>
> ### Weakness 1: Clarification on the Unbiased Gradient Claim
> > W1. Unbiased gradient statement as key contribution is overstated. This property requires taking the expectation over stochastic sampling. Moreover, as the full gradient evolves along the training trajectory, it is unclear to which the step of the full gradient the unbiasedness claim refers. Therefore, the claim of unbiasedness appears to be overstated and lacks rigorous justification.
>
> Thanks for the insightful question. It is indeed the case that unbiasedness can be interpreted differently in the training process. In our paper, the unbiasedness of GUM that we are referring to is stated in Lemma 1. The lemma proves that for **each** step of GUM, for a specific layer, if we consider $\xi$ as the indicator random variable that denotes whether layer $\ell$ is sampled to do a full-rank update, i.e.,
>
> * $\xi = 0$ when layer $\ell$ does a low-rank update, with probability $1-q$
> * $\xi = 1$ when layer $\ell$ does a full-rank update, with probability $q$
>
> Then, after taking expectation over $\xi$, the update of GUM is equivalent to
>
> * $\tilde{M}^+ = \beta \tilde{M} + \tilde{G}$
> * $W^+ = W - \eta \mathrm{NewtonSchulz}(\tilde{M}^+)$
>
> where $\mathbb{E}_\xi[\tilde{G}] = G$ and $G$ denotes the stochastic gradient obtained at the iteration. This means that GUM is an unbiased gradient method, in the sense that it is equivalent to Muon with some additional unbiased noise in gradients. As a reference, note that we say SGD is an unbiased gradient method, it is also in the sense that it is equivalent to gradient descent with some unbiased gradient noise. We hope this explanation can address the concern about why GUM is an unbiased method.
>
> In addition, a general unbiased version of our algorithm is available in Algorithm 3 (Appendix A), where the theoretical guarantees are also provided in Lemma 2 and Lemma B.1 in the Appendix.
>
> We will modify the claim accordingly to make it clearer. Thanks for the suggestion!
>
> ### Weakness 2: Clarification on $q=0.5$ and Full Rankness
> > W2. According to Theorem 1, the optimal choice of q is 0.5, which coincides with the hyperparameter used in the experiments (Line 290). However, using a constant q implies that the full-rank updates dominate the space complexity, matching that of full fine-tuning. This significantly limits the contribution of the work. It would be more meaningful to design an algorithm with a diminishing q, aligning better with the paper’s goal of memory-efficient optimization.
>
> Thanks for the insightful question. We appreciate the reviewer's effort in checking the theorem and pointing out that choosing $q = 0.5$ can lead to the best result.
>
> To provide clarification regarding this concern, we would also like to show that even smaller constant $q$ leads to the same order of convergence rate. This can be proved by applying the same proof in our theoretical guarantee for a small $q$, which we also use in our experiments ($2$ out of $12$ layers using full-rank updates).
>
> The idea of diminishing $q$ can indeed lower the memory cost at the end of training, though a large initial $q$ will still be the bottleneck of the peak memory footprint. It would be an interesting future direction to further consider how to lower the cost of a compensated full-rank update for GUM.
>
> We will incorporate this idea in the discussion of our revisions.
>
> ### Weakness 3: Clarification on the Unbiasedness
> > W3. In Section 5.1, the authors do not use q = 0.5; instead, they only update two layers with full-rank gradients. This further indicates that the proposed method is not an unbiased gradient estimator for low-rank projection under efficient memory constraints, undermining the claimed theoretical properties.
>
> Thanks for the comment. Similar to the response to Weakness 2, any constant $q \in (0,1)$ ensures unbiasedness and the theoretical convergence guarantee of the proposed method, as proved in Lemma 1.
>
> We will provide more clarification in the revision to avoid confusion.

---

> ### Author Response · Authors · 2025-11-21
> **R1: Response to Reviewer SdfL - Part 2**
>
> ### Weakness 4: Longer Training
> > W4. Insufficient training epochs for convergence. For the 8B-parameter model with rank = 512, the reported number of epochs appears insufficient for convergence. The authors are encouraged to extend training to at least 5 epochs to provide a more convincing empirical evaluation.
>
> Thanks for the constructive feedback! To address this concern, we extended the number of epochs and conducted additional experiments on the IFEval benchmark with Gemma-2-9B.
>
> | Epochs    | Prompt-level Strict-Accuracy | Prompt-level Loose-Accuracy |
> | --------  | ------- | ------- |
> | 1 | 33.27   | 36.60    |
> | 3 | 29.57   | 31.42    |
> | 5 | 26.25   | 28.28    |
>
> As shown in the table, the performance degrades with an increasing number of epochs, indicating overfitting. So the number of epochs is sufficient for this supervised fine-tuning setting.
>
> To further investigate GUM's performance in effectively longer training settings, we conducted additional experiments on LLaMA-60M in Table 4, increasing the data amount to 5B tokens (originally 1.5B in Table 4).
>
> | Method    | ARC-E | ARC-C | OBQA | HellaSwag | PIQA | SIQA | Winogrande | Avg. |
> | -------- | ------- | ------- | ------- | ------- | ------- | ------- | ------- | ------- |
> | GaLore | 37.50 | 18.60 | 11.07 | 27.07    |60.66   | 37.56    | 49.57   | 34.58   |
> | Fira  |  37.16 | 17.66 | 14.29 | 27.38    | 60.72    | 37.62   | 50.83    | 35.09    |
> | GUM    | **37.79** | 17.75 | **14.29** | 27.30    | **61.26**    | **37.81**    | **51.67**    | **35.41**    |
>
> As shown in the table above, GUM still outperforms Fira and GaLore by a non-trivial margin, demonstrating the effectiveness of GUM under longer training settings.
>
> ### Question 1: Comparison with Golore
> > Since the authors position GoLore as the closest approach to building an unbiased algorithm, the absence of direct empirical comparison makes the conclusions about landscape properties and convergence rates unconvincing.
>
> Thanks for the insightful suggestion! To address this concern, we have included the additional GoLore baseline in the supervised fine-tuning setting (Table 2) for Gemma-2-9B.
>
> | Method | Prompt-level Strict-Accuracy | Prompt-level Loose-Accuracy | GSM8K |
> | --------  | ------- | ------- | ------- |
> | GaLore    | 30.31   | 33.64   | 77.18   |
> | GoLore    | 31.05   | 34.38   | 74.98   |
> | GUM       | **33.27** | **36.60** | **77.48** |
>
> As shown in the table, GUM still outperforms GaLore by a significant margin in both instruction-following and mathematical reasoning tasks.
>
> ### Question 2: Clarification on the Choice of Muon
> > What is the key innovation in using the optimizer Muon for low-rank adaptation during fine-tuning in this work? The paper does not clearly state why this optimizer provides unique benefits compared to standard choices (e.g., AdamW).
>
> Thanks for the valuable feedback. There are two sides of benefits in employing Muon.
>
> On the empirical side, Muon is demonstrated to perform better than AdamW in pre-training tasks [1][2], as also shown in Table 4. In addition, Muon incurs less memory consumption for common $\sim$7B-sized models since it has no second moments, as also shown in the Gemma-2-9B setting of Table 2. On the theoretical side, Muon satisfies properties I and II in Lemma 2, allowing the unbiasedness to be proven.
>
> We will incorporate the discussion above in the revision of our paper. Thanks for the comment!
>
>
> If you believe our responses have satisfactorily addressed your concerns, we would be most appreciative if you could consider raising the score.
>
> We fully understand the demands on your time and sincerely thank you for your efforts in helping us improve our work. We look forward to any additional feedback you may have.
>
> ### References
>
> [1]: Team, Kimi, et al. "Kimi k2: Open agentic intelligence." arXiv preprint arXiv:2507.20534 (2025).
>
> [2]: Liu, Jingyuan, et al. "Muon is scalable for LLM training." arXiv preprint arXiv:2502.16982 (2025).

---

> ### Author Response · Authors · 2025-11-26
> **Looking forward to further discussion**
>
> Dear Review SdfL,
>
> Thank you for your constructive feedback and insightful suggestions. Your positive remarks about our work are very encouraging, particularly your recognition that it:
>
> * Introduces a new approach for constructing unbiased low-rank optimizers
>
> * Provides solid theoretical convergence analysis of GUM
>
> We are grateful for your assistance in identifying potential concerns in our manuscript, which we have diligently addressed in our response by **incorporating longer training experiments, comparison with Golore, and clarifications on our claims**. With the discussion period deadline approaching, we would greatly appreciate any further comments you may have regarding our response, as we are eager to address any additional issues.
>
> If you find that our responses have adequately addressed your concerns, we would really appreciate it if you could consider raising the score.
>
> We fully understand the demands on your time and sincerely thank you for your efforts in helping us improve our work. We look forward to receiving any additional feedback you may have.

---

> > ### Comment · Reviewer_SdfL · 2025-11-26
> >
> > I acknowledge that I have read the authors’ response. I strongly encourage the authors to mark the revisions in the manuscript (e.g., using a different color) to ensure clear communication during the discussion phase. I maintain my original rating at the moment.

---

> > > ### Author Response · Authors · 2025-11-27
> > > **R2: Response to Reviewer SdfL**
> > >
> > > Dear Review SdfL,
> > >
> > > Thank you so much for your valuable advice and constructive suggestions. To address the mentioned concern, we have included all the results and discussions in our latest manuscript, with the differences highlighted in blue text. In particular, regarding the clarification, we have updated:
> > >
> > > * **The abstract** to clarify the meaning of unbiasedness (Weakness 1)
> > > * **Lemma 1** to rigorously state how the expectation is taken (Weakness 1)
> > > * **Remark 1** to emphasize the fundamental tradeoff between convergence speed and memory consumption when choosing $q$ (Weakness 2, 3)
> > > * **Appendix D.7, Table 11/12** to include results of longer training (Weakness 4)
> > > * **Table 2** to add comparison with GoLore (Question 1)
> > > * **Appendix E.1** to justify our choice of Muon (Question 2)
> > >
> > > If you find that our responses have adequately addressed your concerns, we would really appreciate it if you could consider raising the score.
> > >
> > > We fully understand the demands on your time and sincerely thank you for your efforts in helping us improve our work. We look forward to receiving any additional feedback you may have.

---

### Official Review · Reviewer_9aRW · 2025-10-29

**Soundness:** 3
**Presentation:** 3
**Contribution:** 2
**Rating:** 4
**Confidence:** 2

**Summary:**

In this paper, the authors have introduced GaLore Unbiased with Muon (GUM). It is a new memory efficient algorithm for LLM pretraining and fine-tuning. It is a follow-up work of Galore, where Galore projects gradients into a low-rank subspace to reduce the memory cost. However, one limitation of Galore is that it suffers from a biased gradient estimate. In this paper, the authors address this problem using the layerwise sampling debiasing technique. It performs full-rank updates on some layers and low-rank rank updates on the remaining ones.

**Strengths:**

The strengths of this paper can be summarized as follows:

1. Memory efficiency of LLMs is a very important and popular area of research. The biased gradient updates in low-rank optimizer is a well-known issue. Addressing this is interesting to the machine learning community.

2. The authors have given a rigorous proof for the unbiasedness and convergence. As far as I can see, all the proofs look correct to me.

3. The authors have conducted experiments on both fine-tuning and pre-training on multiple LLMs.

**Weaknesses:**

The weaknesses of this paper are summarized as follows:

1. This paper does not have an ablation study. It would be better to test by changing the sampling probability and projection ranks.

2. Also, the model pretraining mainly focuses on the small models, such as LLaMA-60M, LLaMA-130M, and LLaMA-350M.

3. The improvements are from 0.3% to 1.1%. These improvements are relatively small. The authors may consider showing whether or not these margins are consistent across longer training or larger models.

**Questions:**

Please see weaknesses.

---

> ### Author Response · Authors · 2025-11-21
> **R1: Response to Reviewer 9aRW - Part 1**
>
> Thank you so much for your valuable suggestions and insightful comments. We deeply appreciate your recognition of the significance, theoretical contribution and empirical effectiveness of our work. We would like to address your concerns as follows:
>
> ### Weakness 1: More Ablation Studies
> > This paper does not have an ablation study. It would be better to test by changing the sampling probability and projection ranks.
>
> Thank you for the constructive comment. To address this concern, we have conducted additional ablation studies to better understand the tradeoff between sampling probability, sampling period, and ranks. All experiments are conducted on IFEval benchmark with Gemma-2-9B.
>
> | Number of Sampled Full-Rank Layers    | Prompt-level Strict-Accuracy | Prompt-level Loose-Accuracy |
> | --------  | ------- | ------- |
> | 1 | 30.87   | 34.20   |
> | 2 | 31.61   | 35.12   |
> | 4 | 33.27   | 36.60   |
> | 6 | **33.39**   | **36.75**   |
> | 10| 32.36   | 36.16   |
> | 20    | 28.36   | 30.76    |
>
> As shown in the table above, the best choice of sampled layers is $6$, where the performance starts to degrade when more full-rank layers are introduced. This is consistent with the observation in LISA [1], where this sampling-style training is conjectured to introduce an implicit regularization effect for supervised fine-tuning tasks.
>
> | Sampling Period $K$    | Prompt-level Strict-Accuracy | Prompt-level Loose-Accuracy |
> | --------  | ------- | ------- |
> | 20    | 29.21   | 33.83   |
> | 100   | 30.68   | 35.21   |
> | 200   | **33.27**   | **36.60**  |
> | 500   | 29.39   | 33.27   |
>
> For the best sampling period $K$, our choice of $K=200$ is already optimal, where a smaller $K$ may compromise the momentum and decelerate the training process, while a larger $K$ results in insufficient sampling of all layers.
>
> | Rank $r$    | Prompt-level Strict-Accuracy | Prompt-level Loose-Accuracy |
> | --------  | ------- | ------- |
> | 32    | 30.98   | 36.89   |
> | 64   | 31.24   | 36.60   |
> | 128   | **33.27**   | **36.60**  |
>
> For the best rank, increasing the rank from 32 to 128 leads to overall performance improvements, especially in prompt-level strict accuracy. This means the higher-rank update captures more details for following the given instruction.
>
> ### Weakness 2: Larger-Scale Pre-training
> > Also, the model pretraining mainly focuses on the small models, such as LLaMA-60M, LLaMA-130M, and LLaMA-350M.
>
> Thanks for the valuable feedback. To address this concern, we conducted an additional pre-training experiment on a 7B-sized LLaMA model. Following the setup in [1], we report results using the same number of tokens for pre-training.
>
> | Method    | ARC-E | ARC-C | OBQA | HellaSwag | PIQA | SIQA | Winogrande | Avg. |
> | -------- | ------- | ------- | ------- | ------- | ------- | ------- | ------- | ------- |
> | GaLore | 26.39     | 19.80     | 12.27 | 25.85 | 52.77     | 34.60     | 49.88     | 31.65 |
> | AdamW   | 26.68     | 18.94     | 13.48 | 26.11 | 51.69     | 34.54     | 50.70     | 31.73 |
> | Fira   | 26.46     | 20.90     | 13.01 | 25.65 | 52.34     | 34.75     | 50.54     | 31.95 |
> | GUM    | 26.64 | **20.90** | 12.27 | 25.83 | **53.75** | **35.26** | **50.91** | **32.22** |
>
> As shown in the table, GUM outperforms GaLore, AdamW, and Fira in 7B-sized models as well, with better or no-worse performance on 4 out of 7 tasks, and a higher overall accuracy.

---

> ### Author Response · Authors · 2025-11-21
> **R1: Response to Reviewer 9aRW - Part 2**
>
> ### Weakness 3: Longer Training
> > The improvements are from 0.3% to 1.1%. These improvements are relatively small. The authors may consider showing whether or not these margins are consistent across longer training or larger models.
>
> Thanks for the insightful question. Regarding larger models, according to the 7B-sized pre-training results in our response to Weakness 2 (+0.3% compared with Fira, +0.6% compared with GaLore), this margin is consistent even for larger models.
>
> Regarding longer training settings, we conducted additional experiments on LLaMA-60M in Table 4, increasing the data amount to 5B tokens (originally 1.5B in Table 4).
>
> | Method    | ARC-E | ARC-C | OBQA | HellaSwag | PIQA | SIQA | Winogrande | Avg. |
> | -------- | ------- | ------- | ------- | ------- | ------- | ------- | ------- | ------- |
> | GaLore | 37.50 | 18.60 | 11.07 | 27.07    |60.66   | 37.56    | 49.57   | 34.58   |
> | Fira  |  37.16 | 17.66 | 14.29 | 27.38    | 60.72    | 37.62   | 50.83    | 35.09    |
> | GUM    | **37.79** | 17.75 | **14.29** | 27.30    | **61.26**    | **37.81**    | **51.67**    | **35.41**    |
>
> As shown in the table above, GUM still outperforms Fira and GaLore by a non-trivial margin (+0.3% compared with Fira, +0.8% compared with GaLore), demonstrating the effectiveness of GUM under both scenarios.
>
> If you believe our responses have satisfactorily addressed your concerns, we would be most appreciative if you could consider raising the score.
>
> We fully understand the demands on your time and sincerely thank you for your efforts in helping us improve our work. We look forward to any additional feedback you may have.
>
> ### References
>
> [1]: Rajabi, Sahar, Nayeema Nonta, and Sirisha Rambhatla. "SubTrack++: Gradient Subspace Tracking for Scalable LLM Training." arXiv preprint arXiv:2502.01586 (2025).

---

> ### Author Response · Authors · 2025-11-26
> **Looking forward to further discussion**
>
> Dear Reviewer 9aRW,
>
> We sincerely appreciate your constructive feedback and insightful advice. It is truly encouraging to know that you find our work:
>
> * Shows important limitations of GaLore-style low-rank PEFT methods
>
> * Provides solid theoretical convergence analysis of GUM
>
> * Provides sound evaluation results and demonstrates strong performance
>
> We are grateful for your assistance in identifying potential concerns in our manuscript, which we have diligently addressed in our response by **incorporating more ablation studies, larger-scale pre-training, and longer training experiments**. With the discussion period deadline approaching, we would greatly appreciate any further comments you may have regarding our response, as we are eager to address any additional issues.
>
> If you find that our responses have adequately addressed your concerns, we would really appreciate it if you could consider raising the score.
>
> We fully understand the demands on your time and sincerely thank you for your efforts in helping us improve our work. We look forward to receiving any additional feedback you may have.

---

> > ### Comment · Reviewer_9aRW · 2025-11-26
> >
> > I acknowledge that I have read the responses from the authors. All of my concerns are addressed. I updated my rating.

---

### Official Review · Reviewer_QBWt · 2025-10-31

**Soundness:** 3
**Presentation:** 3
**Contribution:** 3
**Rating:** 6
**Confidence:** 3

**Summary:**

The paper identifies a critical flaw in popular memory efficient optimizers like GaLore: their low-rank gradient projection mechanisms are biased, leading to performance gaps and a lack of convergence guarantees. To solve this, the authors propose a debiasing technique based on layer-wise sampling. Their method, GUM, randomly samples a few layers to perform a compensating full-rank update, while all other layers perform a scaled low-rank update. This ensures the gradient estimate is unbiased in expectation.

**Strengths:**

1. Theoretical Guarantees: It proves that GUM is an unbiased estimator and, as a result, matches the convergence guarantees of its base optimizer (Muon). This directly addresses a major theoretical weakness in GaLore. The synthetic experiment in Figure 1, where GaLore fails to converge but GUM succeeds, provides a stark practical example of this theoretical advantage.

2. Analysis: The paper provides a plausible explanation for GUM's success, linking its high rank updates to a higher stable rank and a more uniform singular value distribution in the model weights, which implies better knowledge distribution and memorization.

3. Performance: GUM consistently outperforms the GaLore baseline in extensive experiments. Fine tuning: On 7B-9B scale models, GUM shows clear improvements over GaLore on both instruction following (IFEval) and reasoning (GSM8K) tasks. Pretraining: On LLaMA models up to 350M, GUM outperforms GaLore on commonsense reasoning benchmarks.

**Weaknesses:**

1. Limited pre-training scale. The pre-training results, while promising, are on relatively small models (up to 350M). Demonstrating the performance win over full rank AdamW on the larger 8B scale models would make the claims even more conclusive.

2. The method introduces new hyperparameters, namely the sampling probability $\gamma$ (or number of full rank layers) and the sampling period $K$. The authors rightly note in their limitations that this sampling can introduce high variance, which leads to instability and requires more careful tuning.

**Questions:**

1. You admit in your limitations that GUM's sampling "introduces high variance," "leads to instability," and "requires more careful tuning". Why is optimizer that's hard to tune a practical improvement over GaLore?
2. In Fig. 4, with (K=200) projector refreshes, the residual $\chi_t$ exceeds ~60–80% within fewer than 20 iterations after each refresh, suggesting severe projector staleness rather than inherent bias. Could this be largely fixed by more frequent refresh? Please report GaLore with smaller (K) (e.g., (K=20)) under matched compute and compare against GUM (quality vs. wall clock/FLOPs).
3. Please report per step FLOPs, wall clock to a fixed validation metric, and SVD costs (including workspace) for GaLore vs. GUM across the same hardware. Provide time vs quality curves.

---

> ### Author Response · Authors · 2025-11-21
> **R1: Response to Reviewer QBWt - Part 1**
>
> Thank you so much for your valuable suggestions and insightful comments. We deeply appreciate your recognition of the theoretical contribution and empirical effectiveness of our work. We would like to address your concerns as follows:
>
> ### Weakness 1: Larger-scale Pre-training
> > Limited pre-training scale. The pre-training results, while promising, are on relatively small models (up to 350M). Demonstrating the performance win over full rank AdamW on the larger 8B scale models would make the claims even more conclusive.
>
> Thanks for the valuable feedback. To address this concern, we conducted an additional pre-training experiment on a 7B-sized LLaMA model. Following the setup in [1], we report results using the same number of tokens for pre-training.
>
> | Method    | ARC-E | ARC-C | OBQA | HellaSwag | PIQA | SIQA | Winogrande | Avg. |
> | -------- | ------- | ------- | ------- | ------- | ------- | ------- | ------- | ------- |
> | GaLore | 26.39   | 19.80     | 12.27 | 25.85 | 52.77     | 34.60     | 49.88     | 31.65 |
> | AdamW   | 26.68  | 18.94     | 13.48 | 26.11 | 51.69     | 34.54     | 50.70     | 31.73 |
> | Fira   | 26.46   | 20.90     | 13.01 | 25.65 | 52.34     | 34.75     | 50.54     | 31.95 |
> | GUM    | 26.64 | **20.90** | 12.27 | 25.83 | **53.75** | **35.26** | **50.91** | **32.22** |
>
> As shown in the table, GUM outperforms GaLore, AdamW, and Fira in 7B-sized models as well, with better or no-worse performance on 4 out of 7 tasks, and a higher overall accuracy.
>
> ### Weakness 2: More Ablation Studies
> > The method introduces new hyperparameters, namely the sampling probability (or number of full rank layers) and the sampling period $K$. The authors rightly note in their limitations that this sampling can introduce high variance, which leads to instability and requires more careful tuning.
>
> Thanks for the constructive comment. To address this concern, we provide additional ablation studies to provide guidelines for hyperparameter tuning, including sampling probability, sampling period, and ranks. All experiments are conducted on IFEval benchmark with IFEval.
>
> | Number of Sampled Full-Rank Layers    | Prompt-level Strict-Accuracy | Prompt-level Loose-Accuracy |
> | --------  | ------- | ------- |
> | 1 | 30.87   | 34.20   |
> | 2 | 31.61   | 35.12   |
> | 4 | 33.27   | 36.60   |
> | 6 | **33.39**   | **36.75**   |
> | 10| 32.36   | 36.16   |
> | 20    | 28.36   | 30.76    |
>
> As shown in the tables above, the best choice of sampled layers is $6$, where the performance starts to degrade when more full-rank layers are introduced. This is consistent with the observation in LISA [1], where this sampling-style training is conjectured to introduce an implicit regularization effect for supervised fine-tuning tasks.
>
> | Sampling Period $K$    | Prompt-level Strict-Accuracy | Prompt-level Loose-Accuracy |
> | --------  | ------- | ------- |
> | 20    | 29.21   | 33.83   |
> | 100   | 30.68   | 35.21   |
> | 200   | **33.27**   | **36.60**  |
> | 500   | 29.39   | 33.27   |
>
> For the best sampling period $K$, our choice of $K=200$ is already optimal, where a smaller $K$ may compromise the momentum and decelerate the training process, while a larger $K$ results in insufficient sampling of all layers.
>
> | Rank $r$    | Prompt-level Strict-Accuracy | Prompt-level Loose-Accuracy |
> | --------  | ------- | ------- |
> | 32    | 30.98   | 36.89   |
> | 64   | 31.24   | 36.60   |
> | 128   | **33.27**   | **36.60**  |
>
> For the best rank, increasing the rank from 32 to 128 leads to overall performance improvements, especially in prompt-level strict accuracy. This means the higher-rank update captures more details for following the given instruction.
>
> These results provide practical guidelines for hyperparameter tuning for $\sim$7B-sized models and ease the effort.

---

> ### Author Response · Authors · 2025-11-21
> **R1: Response to Reviewer QBWt - Part 2**
>
> ### Question 1: Regarding Limitations
> > You admit in your limitations that GUM's sampling "introduces high variance," "leads to instability," and "requires more careful tuning". Why is optimizer that's hard to tune a practical improvement over GaLore?
>
> Thanks for the constructive comment. To ease the hyperparameter tuning effort, we provided further ablation studies, as shown in the response to Weakness 2. Compared to GaLore, GUM resolves its major deficiencies of non-convergence and biasedness, at the cost of higher variance and instability. This variance can be further reduced by introducing variance reduction techniques, such as importance sampling, which we leave for future work.
>
> ### Question 2: GaLore Baseline with small $K=20$
> > In Fig. 4, with (K=200) projector refreshes, the residual $\chi_t$ exceeds ~60–80% within fewer than 20 iterations after each refresh, suggesting severe projector staleness rather than inherent bias. Could this be largely fixed by more frequent refresh? Please report GaLore with smaller (K) (e.g., (K=20)) under matched compute and compare against GUM (quality vs. wall clock/FLOPs).
>
>
> Thanks for the insightful suggestion. To address this concern, we have included an additional comparison between GaLore ($K = 20/200$) and GUM in terms of ARC-E accuracy vs. FLOPS.
>
> As shown in Figure 6 in Appendix D.3 of the latest revision, we observe two main phenomena:
>
> * GUM outperforms GaLore ($K = 20/200$) in terms of computational efficiency.
> * A more frequent projector refresh in GaLore ($K = 20$) does not improve computational efficiency.
>
>
> ### Question 3: Computational Cost Comparison
> > Please report per step FLOPs, wall clock to a fixed validation metric, and SVD costs (including workspace) for GaLore vs. GUM across the same hardware. Provide time vs quality curves.
>
> Thanks for the insightful suggestion. Similar to our response to Question 2, the additional experiments are available in Figure 6 in Appendix D.3.
>
> If you believe our responses have satisfactorily addressed your concerns, we would be most appreciative if you could consider raising the score.
>
> We fully understand the demands on your time and sincerely thank you for your efforts in helping us improve our work. We look forward to any additional feedback you may have.
>
> ### References
>
> [1]: Rajabi, Sahar, Nayeema Nonta, and Sirisha Rambhatla. "SubTrack++: Gradient Subspace Tracking for Scalable LLM Training." arXiv preprint arXiv:2502.01586 (2025).

---

> ### Author Response · Authors · 2025-11-26
> **Looking forward to further discussion**
>
> Dear Reviewer QBWt,
>
> Thank you for your constructive feedback and insightful suggestions. Your positive remarks about our work are very encouraging, particularly your recognition that it:
>
> * Shows important limitations of GaLore-style low-rank PEFT methods
>
> * Provides plausible explanations for GUM's success
>
> * Provides sound evaluation results and demonstrates strong performance
>
> We are grateful for your assistance in identifying potential concerns in our manuscript, which we have diligently addressed in our response by **incorporating larger-scale pre-training experiments, more ablation studies, and computational cost comparisons**. With the discussion period deadline approaching, we would greatly appreciate any further comments you may have regarding our response, as we are eager to address any additional issues.
>
> If you find that our responses have adequately addressed your concerns, we would really appreciate it if you could consider raising the score.
>
> We fully understand the demands on your time and sincerely thank you for your efforts in helping us improve our work. We look forward to receiving any additional feedback you may have.

---

### Official Review · Reviewer_Et17 · 2025-11-01

**Soundness:** 3
**Presentation:** 3
**Contribution:** 2
**Rating:** 2
**Confidence:** 4

**Summary:**

This paper introduces GaLore Unbiased with Muon (GUM), a memory-efficient optimization method for training large language models. GUM debiases low-rank training using layerwise sampling. While prior methods such as GaLore reduce optimizer state memory by projecting gradients into a low-rank subspace, they introduce inherent optimization bias and lack convergence guarantees, leading to degraded or unstable performance, especially under noisy gradients. GUM resolves this by integrating layerwise unbiased sampling with the Muon optimizer, periodically applying full-rank updates to mathematically cancel projection bias in expectation while retaining low-rank efficiency.

**Strengths:**

1. The proposed method highlights and shows an important limitation in GaLore-style low-rank PEFT methods - that most low-rank optimization methods introduce biased gradient estimations during training. This paper also rightly highlights the challenges in analyzing the convergence properties of such methods.
2. Theoretical Contribution and convergence analysis of GUM is an important contribution. The motivating example to show why GaLore might fail in an extremely noisy setting highlights an important issue.

**Weaknesses:**

1. Some ablations on the probability of full-rank updates, sampling period, or rank of low-rank updates could be insightful.
2. Inherent issues with Muon as an optimizer haven’t been fixed or addressed
- i) for large hidden layers, computational overhead from Newton-Schulz Updates would be significant;
- ii) Muon has only been studied for dense hidden linear layers and stability and efficiency will degrade in a sparse training regime;
- iii) Although Muon improves conditioning without second moments, storing intermediate matrices and running iterative updates might still use more memory than other optimizers.
- iv) how does Muon perform for larger LLMs (30B/70B scale)?

3. Experiments lack a performance comparison to other low-rank SOTA methods, especially the ones that could be potentially unbiased due to their recovery, random selection, or error feedback mechanisms, like: 1) LDAdam [1], 2) APOLLO [2], 3) GreedeLore [3] 4) FRUGAL [4] 5) SubTrack++ [5], etc.
- i) Pre-training experiments are conducted on very small models, and can not be used to show the effectiveness of proposed methods in larger (7B+), and longer training regimes.
- ii) Although Fira is included in pre-training baselines as one potentially unbiased low-rank method; it is not SOTA anymore, and methods like LDAdam [1], APOLLO [2], or SubTrack++ [5] has been shown to surpass its performance.
- iii) None of the mentioned unbiased low-rank techniques are included in fine-tuning experiments for a fair comparison and thorough evaluation of the proposed method.

 [1] Robert et al., 2025. LDAdam: Adaptive Optimization from Low-Dimensional Gradient Statistics.

 [2] Zhu et al., 2025. APOLLO: SGD-like Memory, AdamW-level Performance.

 [3] Chen et al., 2025. Greedy Low-Rank Gradient Compression for Distributed Learning with Convergence Guarantees.

[4] Zmushko et al., 2024. FRUGAL: Memory-Efficient Optimization by Reducing State Overhead for Scalable Training.

[5] Rajabi et al., 2025. SubTrack++: Gradient Subspace Tracking for Scalable LLM Training.

**Questions:**

Please refer to the weaknesses.

---

> ### Author Response · Authors · 2025-11-21
> **R1: Response to Reviewer Et17 - Part 1**
>
> Thank you so much for your valuable suggestions and insightful comments. We deeply appreciate your recognition of the significance and theoretical contribution of our work. We would like to address your concerns as follows:
>
> ### Weakness 1: More Ablation Studies
> > Some ablations on the probability of full-rank updates, sampling period, or rank of low-rank updates could be insightful.
>
> Thank you for the constructive comment. To address this concern, we have conducted additional ablation studies to better understand the tradeoff between sampling probability, sampling period, and ranks. All experiments are conducted on IFEval benchmark with Gemma-2-9B.
>
> | Number of Sampled Full-Rank Layers | Prompt-level Strict-Accuracy | Prompt-level Loose-Accuracy |
> | --------  | ------- | ------- |
> | 1 | 30.87   | 34.20   |
> | 2 | 31.61   | 35.12   |
> | 4 | 33.27   | 36.60   |
> | 6 | **33.39**   | **36.75**   |
> | 10| 32.36   | 36.16   |
> | 20    | 28.36   | 30.76    |
>
> As shown in the table above, the best choice of sampled layers is $6$, where the performance starts to degrade when more full-rank layers are introduced. This is consistent with the observation in LISA [1], where this sampling-style training is conjectured to introduce an implicit regularization effect for supervised fine-tuning tasks.
>
> | Sampling Period $K$    | Prompt-level Strict-Accuracy | Prompt-level Loose-Accuracy |
> | --------  | ------- | ------- |
> | 20    | 29.21   | 33.83   |
> | 100   | 30.68   | 35.21   |
> | 200   | **33.27**   | **36.60**  |
> | 500   | 29.39   | 33.27   |
>
> For the best sampling period $K$, our choice of $K=200$ is already optimal, where a smaller $K$ may compromise the momentum and decelerate the training process, while a larger $K$ results in insufficient sampling of all layers.
>
> | Rank $r$   | Prompt-level Strict-Accuracy | Prompt-level Loose-Accuracy |
> | --------  | ------- | ------- |
> | 32    | 30.98   | **36.89**   |
> | 64   | 31.24   | 36.60   |
> | 128   | **33.27**   | 36.60  |
>
> For the best rank, increasing the rank from 32 to 128 leads to overall performance improvements, especially in prompt-level strict accuracy. This means the higher-rank update captures more details for following the given instruction.
>
> ### Weakness 2: Clarification on Muon
> > Inherent issues with Muon as an optimizer haven’t been fixed or addressed.
> > i) for large hidden layers, computational overhead from Newton-Schulz Updates would be significant;
> > ii) Muon has only been studied for dense hidden linear layers and stability and efficiency will degrade in a sparse training regime;
> > iii) Although Muon improves conditioning without second moments, storing intermediate matrices and running iterative updates might still use more memory than other optimizers.
> > iv) how does Muon perform for larger LLMs (30B/70B scale)?
>
> Thanks for the comment. The main focus of our algorithm is not the Muon optimizer itself, but a technique for debiasing existing low-rank training methods like GaLore, which is empirically orthogonal to the underlying optimizers such as AdamW and Muon.
>
> But we are happy to clarify more details about Muon to address the concerns mentioned.
>
> * i) There is a fundamental tradeoff between Muon and AdamW across different model sizes.
>     - Generally, Muon favors **deep and thin** networks, while AdamW has memory advantages in **large-scale wide** networks. On one hand, Muon may incur higher memory cost for extremely large hidden layers, since Muon requires matrix–matrix multiplication in the Newton–Schulz5 update, whereas AdamW only requires matrix–vector multiplication operations. On the other hand, Muon has only one momentum term, while AdamW has an additional second moment, which incurs extra memory consumption.
>     - For commonly used $\sim$7B-sized models like Gemma-2-9B, AdamW empirically requires more GPU memory, as shown in Table 2 of our paper, where AdamW triggers an out-of-memory error while Muon does not.
>
> * ii) This is not true. Muon has been successfully applied to Mixture-of-Experts training [2] and outperforms AdamW, as shown in Table 4 of [3].
>
> * iii) See (i).
>
> * iv) Muon performs well in large LLMs, as demonstrated by Kimi K2 [2], which has 1T total parameters and 32B activated parameters.

---

> ### Author Response · Authors · 2025-11-21
> **R1: Response to Reviewer Et17 - Part 2**
>
> ### Weakness 3.1: More Baselines
> > Experiments lack a performance comparison to other low-rank SOTA methods, especially the ones that could be potentially unbiased due to their recovery, random selection, or error feedback mechanisms, like: 1) LDAdam [1], 2) APOLLO [2], 3) GreedeLore [3] 4) FRUGAL [4] 5) SubTrack++ [5], etc.
> > ii) Although Fira is included in pre-training baselines as one potentially unbiased low-rank method; it is not SOTA anymore, and methods like LDAdam [1], APOLLO [2], or SubTrack++ [5] has been shown to surpass its performance.
> > iii) None of the mentioned unbiased low-rank techniques are included in fine-tuning experiments for a fair comparison and thorough evaluation of the proposed method.
>
> Thanks for the insightful question. To address the mentioned concern, we have included comparisons with LDAdamW, Apollo, and SubTrack++ on IFEval and GSM8K using the Qwen-2.5-7B model.
>
> | Method | Prompt-level Strict-Accuracy | Prompt-level Loose-Accuracy | GSM8K Acc.|
> | --------  | ------- | ------- |  ----|
> | LDAdamW    | 28.10   | 30.31    | 83.78 |
> | Apollo     | 31.61   | 36.41    | 85.67 |
> | SubTrack++ | 29.76   | 34.01    | 86.66 |
> | GUM        | **33.46**   | **38.82**    |  **86.81** |
>
> As shown in the tables above, GUM outperforms all the baselines on both IFEval and GSM8K under the same memory budget constraint. SubTrack++ is by far the strongest baseline in reasoning tasks, achieving performance close to GUM on GSM8K. We will include all of these results in the revision of our paper.
>
> ### Weakness 3.2: Larger-scale Pre-training
> > Pre-training experiments are conducted on very small models, and can not be used to show the effectiveness of proposed methods in larger (7B+), and longer training regimes.
>
> Thanks for the valuable feedback. To address this concern, we conducted an additional pre-training experiment on a 7B-sized LLaMA model. Following the setup in SubTrack++ [4], we report results using the same number of tokens for pre-training.
>
> | Method    | ARC-E | ARC-C | OBQA | HellaSwag | PIQA | SIQA | Winogrande | Avg. |
> | -------- | ------- | ------- | ------- | ------- | ------- | ------- | ------- | ------- |
> | GaLore | 26.39     | 19.80     | 12.27 | 25.85 | 52.77     | 34.60     | 49.88     | 31.65 |
> | AdamW   | 26.68     | 18.94     | 13.48 | 26.11 | 51.69     | 34.54     | 50.70     | 31.73 |
> | Fira   | 26.46     | 20.90     | 13.01 | 25.65 | 52.34     | 34.75     | 50.54     | 31.95 |
> | GUM    | 26.64 | **20.90** | 12.27 | 25.83 | **53.75** | **35.26** | **50.91** | **32.22** |
>
> As shown in the table, GUM outperforms GaLore, AdamW, and Fira in 7B-sized models as well, with better or no-worse performance on 4 out of 7 tasks, and a higher overall accuracy.
>
>
> If you believe our responses have satisfactorily addressed your concerns, we would be most appreciative if you could consider raising the score.
>
> We fully understand the demands on your time and sincerely thank you for your efforts in helping us improve our work. We look forward to any additional feedback you may have.
>
> ### Reference
>
> [1]: Pan, Rui, et al. "Lisa: Layerwise importance sampling for memory-efficient large language model fine-tuning." NeurIPS 2024.
>
> [2]: Team, Kimi, et al. "Kimi k2: Open agentic intelligence." arXiv preprint arXiv:2507.20534 (2025).
>
> [3]: Liu, Jingyuan, et al. "Muon is scalable for LLM training." arXiv preprint arXiv:2502.16982 (2025).
>
> [4]: Rajabi, Sahar, Nayeema Nonta, and Sirisha Rambhatla. "SubTrack++: Gradient Subspace Tracking for Scalable LLM Training." arXiv preprint arXiv:2502.01586 (2025).

---

> ### Author Response · Authors · 2025-11-26
> **Looking forward to further discussion**
>
> Dear Reviewer Et17,
>
> We sincerely appreciate the time and effort you have invested in reviewing our manuscript and providing such insightful suggestions and comments. We are pleased that you find our paper:
>
> * Shows important limitation of GaLore-style low-rank PEFT methods
>
> * Provides solid theoretical convergence analysis of GUM
>
> We are grateful for your assistance in identifying potential concerns in our manuscript, which we have diligently addressed in our response by **incorporating more ablation studies, baselines, and larger-scale pre-training experiments**. With the discussion period deadline approaching, we would greatly appreciate any further comments you may have regarding our response, as we are eager to address any additional issues.
>
> If you find that our responses have adequately addressed your concerns, we would really appreciate it if you could consider raising the score.
>
> We fully understand the demands on your time and sincerely thank you for your efforts in helping us improve our work. We look forward to receiving any additional feedback you may have.

---

### Author Response · Authors · 2025-12-03
**Rebuttal summary**

Dear SAC, AC, and Reviewers,

We sincerely appreciate the time and effort you have invested in reviewing our manuscript and providing such insightful suggestions and comments. We are pleased to note that the reviewers acknowledged that our paper:

* **Shows important limitation of GaLore-style low-rank PEFT methods** (Et17, QBWt, 9aRW, SdfL)
* **Provides solid theoretical convergence analysis of GUM** (Et17, QBWt, 9aRW, SdfL)
* **Provides sound evaluation results and demonstrate strong performance** (QBWt, 9aRW)

To save time for every reviewer, AC, and SAC, we would like to share a brief summary of the rebuttal and the relevant concerns raised during the discussion.

* Updated ratings
  - Reviewer QBWt: 6
  - Reviewer SdfL: 6
  - Reviewer 9aRW: 4 $\rightarrow$ 6 (before the Nov. 27 incident)
  - Reviewer Et17: 2 (unable to respond because of the system lock)

* Relevant Concerns (**all addressed in the updated manuscript**):
|Concern| Addressed in | Raised by Reviewer |
|-|-|-|
|More baselines | Table 4, Table 5, and Appendix D.5 | Et17, QBWt, SdfL |
|Computational Cost Comparison | Figure 6 in Appendix D.3 | QBWt |
|More ablation studies | Table 8 in Appendix D.4 | Et17, QBWt, 9aRW |
| Larger-scale pre-training (~7B model) | Table 10 in Appendix D.6 | Et17, QBWt, 9aRW |
| Longer training | Table 11 & 12 in Appendix D.7 | 9aRW, SdfL |
| Clarification on Muon | Appendix E.1 | Et17, SdfL |
| Clarification on Unbiased Gradient | Abstract, Lemma 1 and Remark 1 | SdfL |

Thank you very much for reviewing the comments, and we hope this summary can help reduce the workload.

Sincerely,

Paper 19147 Authors

---

### Meta-Review · Area_Chair_P7C7 · 2026-01-06

**Summary:**

The paper propose GUM, which use layer-wise sampling to fix bias in low-rank optimization for large language models. It prove convergence and show better memory efficiency and performance than GaLore on big models.

There have been many strengths of the paper acknowledged by the reviewers, but that major concerns that the reviewers highlighted are listed below:

1. the claim of "unbiased gradient" is overstated because it only true when taking expectation over many steps. The reviewer also note that using a high sampling probability ($q=0.5$) might make memory usage same as full training
2. many reviewers feel that pre-training only on small models like LLaMA-350M is not enough to prove method works for real LLMs
3. many new low-rank methods are missing for comparison, specifically those that use random selection or error feedback like 1) LDAdam [1], 2) APOLLO [2], 3) GreedeLore [3] 4) FRUGAL [4] 5) SubTrack++ [5], etc].
4. some reviewers are worried about the new hyperparameters like sampling period $K$ and probability $q$ .
5. there are several technical problems with using Muon, such as high overhead for large layers and lack of testing on sparse training

**Reviewer Concerns:**

# Addressed Concerns

I feel that authors did a very good job to addressing the reviewers concerns, namely
1. they successfully address the experimental scale issue by adding new pre-training results on a 7B-sized LLaMA model. The results show GUM still better than AdamW and GaLore at this bigger scale
2.  Missing SOTA Baselines is now mostly resolved. Authors include new comparisons with LDAdamW, Apollo, and SubTrack++. GUM show better performance than these strong baselines
3. Authors provide very detailed ablation tables for the sampling period $K$, rank, and number of layers. These tables give clear guidelines for tuning on 7B models
4. Authors clarify that Muon is actually very efficient for 7B models because it skip the second moment that AdamW needs. They also point to other papers where Muon works for sparse Mixture-of-Experts


# Outstanding Concerns
*  While authors explain that GUM is unbiased in expectation, some Reviewers still seem not fully convinced.  The worry about $q=0.5$ making memory high is partially addressed by authors saying they only use 2 or 4 layers in practice, but the theoretical tension between high $q$ for convergence and low $q$ for memory still exist.
*  for the suggested 5 SOTA baselines where some also try to fix GaLore bias authors added results for LDAdamW, Apollo, and SubTrack++ and show GUM is better. However, GreedeLore [3] and FRUGAL [4] were still not tested in the rebuttal. This mean we don't know for sure if GUM is better than every unbiased method suggested (and authors could just cherry-picked the one where GUM is superior).

**Reviewer Scores:**

*  Reviewer Et17 (Initial: 2) - I think many of the reject reasons have been addressed, but not all. I think he would increase to 4 (GreedeLore and FRUGAL were still not tested) or 6.
*  Reviewer SdfL (Initial: 6) - I feel that he would keep his score
*  Reviewer 9aRW (Initial: 4) - I feel that many issues have been addressed and he would end up with score of 6.
*  Reviewer QBWt (Initial: 6)  - I feel his concerns have been addressed quite well and he would most likely increase his score to 8.

---

### Decision · Program_Chairs · 2026-01-26

Reject